# Dual-Scale World Memory for LLM Agents towards Hard-Exploration Problems

**Minsoo Kim**   **Seung-won Hwang**[*]
Seoul National University
{`minsoo9574, seungwonh`}@snu.ac.kr

## Abstract

LLM-based agents have seen promising advances, yet are still limited in *hard-exploration* tasks which require agents to perform sustained exploration under sparse feedback. We present GLoW, a novel approach leveraging a dual-scale textual world memory, maintaining a trajectory frontier of high-value discoveries at the global scale, while learning from local trial-and-error in exploration through a Multi-path Advantage Reflection mechanism which infers advantage-based progress signals to guide exploration. To evaluate our framework for hard-exploration, we tackle the Jericho benchmark suite of text-based games, where GLoW achieves a new state-of-the-art performance for LLM-based approaches. Compared to state-of-the-art RL-based methods, our approach achieves comparable performance while requiring 100-800× fewer environment interactions. When scaled to stronger LLMs, GLoW surpasses all prior methods on 4 out of 6 difficult and extreme Jericho games.[1]

## 1 Introduction

While LLM agents (Yao et al., 2023; Sumers et al., 2024; Wang et al., 2024) excel at leveraging vast pre-trained knowledge in tasks such as robotic planning, software engineering, and web automation (Ahn et al., 2022; Yang et al., 2024; 2025), they are reportedly limited in *hard-exploration problems* (Sutton & Barto, 2018; Ecoffet et al., 2021). Hard exploration problems are typically characterized by large state–action spaces, deceptive local optima, and sparse rewards. These factors often trap naive exploration in local optima, such that exploration fails to reach deeper states with rewards. For LLM agents, such problems pose two central challenges: (1) Global learning, for maintaining long-term knowledge of valuable discoveries during exploration, (2) Local trial-and-error, for quickly refining exploration policies from sparse environmental feedback. Current LLM agent approaches such as ReAct (Yao et al., 2023) or Reflexion (Shinn et al., 2023) support local trial-and-error, but lack mechanisms for long-term knowledge accumulation. Consequently, LLM agents fall short on hard-exploration tasks that humans can often solve effectively (Cui et al., 2025; Phan et al., 2025).

In this work, we introduce **G**lobal-**Lo**cal **W**orld Memory (GLoW), a framework enabling effective exploration in hard-exploration problems through dual-scale *world memory* for global and local learning. Inspired by recent views of LLM world models as building implicit representations of task-relevant world knowledge (Li et al., 2024; Ding et al., 2025), our world memory modules capture experiential knowledge from exploration trajectories as structured textual representations, rather than modeling transition dynamics. To instantiate our dual-scale world memory, we build on the Go-Explore (Ecoffet et al., 2021) algorithm, which achieves breakthroughs on hard-exploration problems by enhancing the exploration capabilities of RL and LLM-based agents (Lu et al., 2025). The key idea of Go-Explore is to store discovered states into a *state archive*. Then, based on this archive, Go-Explore decomposes hard-exploration into alternating between: (1) a *selection* phase, choosing a *promising* state from the archive to return to, and (2) an *exploration* phase, to continue discovering new states from the selected state. In its original implementation, Go-Explore used

---

[*]Corresponding author.
[1]Code is available at `https://github.com/mnskim/glow`

hand-crafted heuristics for selection, and random action sampling for exploration, while later work, such as IGE (Lu et al., 2025) improved selection to leverage LLM inference.

In this work, our core insight is that both selection and exploration require structured learning from past exploration experiences, but at different scales: we first enrich beyond an archive of isolated states, by additionally maintaining a trajectory frontier, which keeps the full temporal context of how high value states were reached and why progress stalled, into a **global world memory** for richer structured learning. This allows an LLM-based analysis across the frontier to infer high-value regions as well as bottleneck states with high future potential, enabling principled state selection beyond heuristic or LLM-internalized notions of interestingness. At the local scale, to guide exploration from the chosen state, we draw insights that advantage-based rewards better capture progress signals than Q-values (Kazemnejad et al., 2025; Setlur et al., 2025): Our Multi-path Advantage Reflection mechanism explores multiple trajectories from the same starting state and leverages LLM reasoning to infer *intermediate advantages at key state-action pairs*. Through these advantage signals, the **local world memory** enables controlled exploration under sparse environmental feedback.

To evaluate the capability of LLM agents in hard-exploration problems, we study the Jericho benchmark suite of text-based games (Hausknecht et al., 2020), where state-of-the-art methods have been RL-based solutions with $\varepsilon$-greedy or softmax exploration (Hausknecht et al., 2020; Ammanabrolu & Hausknecht, 2020; Guo et al., 2020), and MCTS-based exploration (Jang et al., 2021; Shi et al., 2025). However, these methods suffer from poor sample efficiency, relying on extensive trial-and-error which requires **hundreds of thousands** of environment interactions. Meanwhile, existing LLM agents have been insufficient to address the challenge of learning from exploration in Jericho games, showing limited performance compared to humans (Cui et al., 2025; Phan et al., 2025).

Through extensive experiments, we show that GLoW improves the performance of LLM-based agents while achieving orders of magnitude improvement in sample efficiency compared to RL baselines. Our contributions are summarized as follows:

- We propose GLoW, a novel LLM agent framework for hard-exploration problems through global-local world memory.

- We conduct comprehensive comparisons with existing agent approaches (RL, MCTS, LLM) and ablation studies to validate the components of our method.

- We achieve a new state of the art for LLM-based approaches on Jericho, achieving performance comparable to the best RL-based methods while reducing environment interactions required by 100-800×.

## 2 BACKGROUND

**Jericho Benchmark** The Jericho benchmark (Hausknecht et al., 2020) remains an unsolved hard-exploration problem, where the text-based game environments provide two fundamental challenges (Ammanabrolu & Riedl, 2021): (1) partial observability, requiring agents to construct models of the world from local textual descriptions, and (2) combinatorial state-action spaces. For example in Zork1, the game vocabulary has 697 words and up to five-word commands, resulting in $O(697^5) = 1.64 \times 10^{14}$ possible actions per step, though only a tiny fraction are grammatically coherent and contextually relevant. As a result, RL approaches, with simple exploration strategies, incur hundreds of thousands interactions to offset sample inefficiencies in exploration. This makes Jericho an ideal testbed for evaluating whether agents learn by exploring, rather than brute-force discovery.

**Methods for Hard-Exploration Problems** Go-Explore (Ecoffet et al., 2021) achieved breakthroughs in hard-exploration problems by maintaining an archive of discovered states as global knowledge to (1) *select* promising states and (2) *explore* from the state. Algorithm 1 illustrates this framework across Go-Explore variants, contrasting selection and exploration strategies. The original algorithm uses novelty-based heuristics for state selection and random actions for exploration. XTX (Tuyls et al., 2022) improves upon these with imitation learning for selection and DQN with curiosity rewards for exploration, while IGE (Lu et al., 2025) leverages LLM inference for both phases. Our approach introduces two key innovations: (1) a trajectory frontier $\mathcal{F}$ with LLM-based value decomposition for principled state selection, and (2) Multi-path Advantage Reflection (MAR)

---

**Algorithm 1** Go-Explore-based Algorithms

---

1: **procedure** GO-EXPLORE-FAMILY($s_0, n_{iter}$)
2:     $\mathcal{A} \leftarrow \{(s_0, 0)\}$                               ▷ Archive of (state, score)
3:     $\mathcal{T} \leftarrow \emptyset$                                  ▷ Collected trajectories
4:     $\mathcal{F} \leftarrow \emptyset$                                  ▷ Trajectory Frontier
5:     **for** $i = 1$ to $n_{iter}$ **do**
        *— Go Phase (State Selection) —*
6:         **Go-Explore:** $s_{next} \sim h(\mathcal{A})$     ▷ Hand-crafted heuristic (e.g., visit count, domain score)
7:         **XTX:** $s_{next} \leftarrow$ ImitationLearning($\mathcal{T}$)                ▷ Imitation learning
8:         **IGE:** $s_{next} \leftarrow$ LLM.SelectPromising($\mathcal{A}$)         ▷ Ill-defined promising-ness
9:         **GLoW:** $W_{\text{global}} \leftarrow g_{\text{LLM}}(\mathcal{F})$        ▷ Principled value decomposition (Sec. 3.1)
10:             $s_{next} \leftarrow$ align$_{\text{LLM}}(\mathcal{A}, W_{\text{global}})$
11:

        *— Explore Phase —*
12:         **Go-Explore:** $\tau \leftarrow$ RandomActions($s_{next}$)              ▷ No learning
13:         **XTX:** $\tau \leftarrow$ DQN($s_{next}$)             ▷ DQN with curiosity reward
14:         **IGE:** $\tau \leftarrow$ ReAct($s_{next}$)           ▷ Standard LLM agent
15:         **GLoW:**
16:         **for** $j = 1$ to $n$ **do**       ▷ LLM agent with advantage-driven exploration (Sec. 3.2)
17:             $\tau_j \leftarrow \pi_{explore}(s_{next}, W_{\text{local}}, \{\tau_1, \ldots, \tau_{j-1}\}, \mathcal{F})$
18:             $W_{\text{local}} \leftarrow$ MAR($\{\tau_1, \ldots, \tau_j\}, \mathcal{F}$)
19:         **end for**
20:         $\mathcal{T} \leftarrow \mathcal{T} \cup \{\tau_1, \ldots, \tau_n\}$            ▷ Collect trajectories
21:         $\mathcal{F} \leftarrow$ top-$k(\mathcal{F} \cup \{\tau_1, \ldots, \tau_n\}, v)$        ▷ Update trajectory frontier
22:

        *— Archive Update —*
23:         **for** each state $s'$ in $\tau$ **do**
24:             **if** IsNotRedundant($s', \mathcal{A}$) **then**         ▷ Domain-specific novelty
25:                $\mathcal{A} \leftarrow \mathcal{A} \cup \{s'\}$
26:             **end if**
27:         **end for**
28:     **end for**
29: **end procedure**

---

for learning from local exploration. Beyond the Go-Explore family, MCTS-based methods like MC-LAVE (Jang et al., 2021) and MC-DML (Shi et al., 2025) leverage tree search with language-driven exploration and LLM priors respectively, though requiring 400,000+ interactions.

## 3 METHOD

In this section, we describe the dual-scale learning paradigm of GLoW's world memory.

### 3.1 GLOBAL WORLD MEMORY FOR STATE SELECTION

The global world memory module extracts value signals from accumulated exploration trajectories. Unlike traditional state-based archives, we maintain trajectories in a value-ranked frontier. The global world memory module additionally maintains LLM-generated trajectory analysis.

**Value-Ranked Trajectory Frontier** As the source of value information, the global world memory module maintains a trajectory frontier $\mathcal{F} = \{\tau_1, \tau_2, ..., \tau_k\}$, containing the $k$ highest-value trajectories discovered during exploration, ranked by a value function $v : \mathcal{T} \rightarrow \mathbb{R}$. Each trajectory $\tau_i = (s_0^i, a_1^i, r_1^i, s_1^i, ..., a_T^i, r_T^i, s_T^i)$ represents a complete episode generated by the exploration policy $\pi_{\text{explore}}$ defined by the LLM agent, where $s_t \in \mathcal{S}$ are states, $a_t \in \mathcal{A}$ are actions, and $r_t \in \mathbb{R}$ are rewards. For the trajectory value function $v$, we use the maximum cumulative reward achieved during the episode, $v(\tau_i) = \max_{t \in [1,T]} \sum_{j=1}^{t} r_j^i$. This is an effective choice for Jericho's sparse reward structure, where agents can encounter negative rewards or terminal failures. In contrast to state-only representations, which lose the context of action and observation sequences, preserving complete trajectories enables accurate credit assignment and value estimation in sparse-reward environments where success depends on precise action sequences. For instance, in Zork1, progressing past the

troll requires first acquiring both the lantern and the sword before descending into the cellar, but only entering the cellar yields a reward. Analyzing complete trajectories, which may each capture different portions of these sequential dependencies, enables inferring across them that both items are necessary despite the sparse feedback.

The frontier evolves progressively through iterative exploration. When exploration from selected states (detailed in Section 3.2) produces trajectory $\tau_{\text{new}}$ with value $v(\tau_{\text{new}})$, the frontier is updated:

$$\mathcal{F}_{t+1} = \text{top-}k(\mathcal{F}_t \cup \{\tau_{\text{new}}\}, v) \tag{1}$$

This sliding window mechanism ensures the frontier maintains diverse high-value strategies, while allowing newly discovered superior trajectories to replace outdated ones. For any state $s_i$, we can derive the achieved value $v(s_i) = \max_{\tau \in \mathcal{F}, s_i \in \tau} v(\tau)$, representing the maximum value reached from state $s_i$ across all frontier trajectories. By tracking complete trajectories, the frontier serves as both an estimator of achieved values and a repository of successful action sequences.

**Motivation: Decomposing value for *select* and *explore*** Inspired by UCB's value decomposition which balances exploitation with exploration bonus as:

$$\bar{V}(s) + c\sqrt{\frac{\log(N)}{n_s}}$$

where $\bar{V}(s)$ is the empirical mean value and the second term is the exploration bonus based on visit count $n_s$, we annotate two types of values $v$ and $v'$, corresponding to each term, by analyzing patterns across all frontier trajectories $\mathcal{F}$, to extract a set of critical global states with value annotations, forming the global world memory:

$$W_{\text{global}} = g_{\text{LLM}}(\mathcal{F}) = \{(s_1, v_1, v_1'), (s_2, v_2, v_2'), \ldots, (s_k, v_k, v_k')\} \tag{2}$$

Here, each $(s_i, v_i, v_i')$ represents a critical global state, key semantic landmarks such as exploration frontiers, bottlenecks, and milestones, identified from frontier analysis by a prompted LLM $g_{LLM}$, where $v_i$ denotes the achieved value from $s_i$, while $v_i'$ reflects LLM's estimate of future value potential. Importantly, this potential value $v_i'$ cannot be derived from trajectory scores alone, requiring LLM's reasoning about why trajectories fail and what progress could be achieved by resolving current bottlenecks. For instance, a state where multiple trajectories fail might have *low achieved value*, but have *high potential value* when: (1) multiple high-value trajectories converge but fail to progress further, suggesting unexplored regions beyond, (2) partial solution patterns indicate missing components, or (3) environmental hints suggest valuable areas remain undiscovered. This implements a *semantic* form of optimism under uncertainty (Auer, 2003; Brafman & Tennenholtz, 2003) where UCB uses statistical bonuses while we derive optimistic values from LLM analysis of bottlenecks. See Appendix F.1 for a full example of $W_{global}$ generated for Zork1.

**Balancing Exploitation and Exploration in State Selection** We maintain a state archive $\mathcal{A} = \{(s_i, \text{score}(s_i))\}$ containing discovered states with their achieved scores. Given $W_{global}$, we select the next exploration state $s_{next}$ by balancing achieved and potential values via LLM as shown in Fig. 1-(a). We leverage $\text{align}_{\text{LLM}}$, an LLM-based state selection operation which evaluates how

**Algorithm 2** Value-aligned State Selection

**Require:** Frontier $\mathcal{F}$, State archive $\mathcal{A}$
**Ensure:** Selected state $s_{next}$
1: $W_{global} \leftarrow g_{LLM}(\mathcal{F})$ where
   $W_{global} = \{(s_1, v_1, v_1'), \ldots, (s_k, v_k, v_k')\}$
   $v_i$: achieved, $v_i'$: potential
2: **for** each state $s \in \mathcal{A}$ **do**
3:     $score[s] \leftarrow align_{LLM}(s, W_{global})$
4: **end for**
5: $s_{next} \leftarrow \arg\max_{s \in \mathcal{A}} score[s]$
6: **return** $s_{next}$

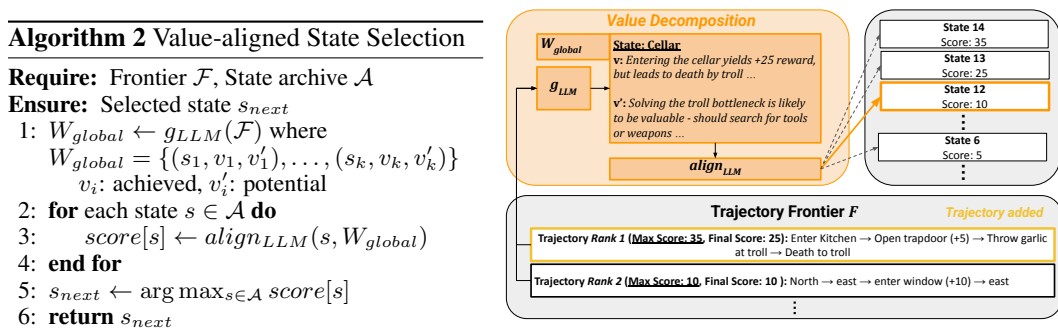

Figure 1: (a) Select procedure in GLoW, (b) Illustration of selection with Global World Memory

well each archived state $s$ aligns with the high-value patterns identified in $W_{\text{global}}$ using a prompted

LLM (see Appendix E.2 for the full prompt). Since $W_{\text{global}}$ contains both achieved and potential values for key frontier states, this alignment naturally balances exploitation (favoring states similar to proven high-reward regions), with exploration (prioritizing states near identified bottlenecks with high potential). Fig. 1-(b) illustrates selection in GLoW with the Global World Memory Module where a new trajectory (highlighted in gold) has been added to the frontier. Once a state is chosen, we replay the stored sequence of actions to return to the state[2], which becomes the starting point of the next exploration phase, described in the following section.

### 3.2 LOCAL WORLD MEMORY FOR EXPLORATION

In addition to the selection of states which align with exploration goals with high potential value, exploration can be enhanced by learning which actions are likely to lead to further progress, which is the objective of the local world memory module.

**Motivation: From Q-values to Advantages for Exploration** Existing LLM learning methods like self-reflection can be viewed as estimating state-action values (Q-values) from single trajectories. However, Q-value estimation from sparse rewards is notoriously high-variance (Sutton et al., 1999; Schulman et al., 2017), and we observe the same challenge in LLM-based learning: inferences from entire trajectories with sparse feedback are prone to incorrect causal attribution.

Drawing from RL theory, advantage functions $A(s, a) = Q(s, a) - V(s)$ reduce variance by comparing actions to a baseline rather than estimating absolute values. Recent work on process reward models (PRMs) further demonstrates that advantage-based rewards are more suited for exploration, by better capturing progress signals than Q-values, which tend to exploit known strategies (Setlur et al., 2025; Kazemnejad et al., 2025).

**Multi-path Advantage Reflection (MAR)** Inspired by TRPO (Schulman et al., 2015), which computes robust advantage in sparse-reward setting over multiple rollouts from the same state, we propose Multi-path Advantage Reflection to compare multiple trajectories from the same starting state, to produce pseudo-dense advantage signals from sparse environmental feedback. This effectively densifies the reward signal by inferring intermediate advantages at key state-action pairs, providing rich guidance for exploration where environmental rewards are insufficient.

Given a state $s$ selected based on the global world memory, we perform iterative exploration by sampling $n$ trajectories sequentially: after each trajectory $\tau_i$, we perform MAR to extract learnings that inform the next trajectory $\tau_{i+1}$, in the form of local world memory $W_{local}$. This creates a sequence $\mathcal{T}_s = \{\tau_1, \tau_2, ..., \tau_n\}$ where each trajectory benefits from insights gained from previous attempts.

---

**Algorithm 3** Exploration with MAR

**Require:** Selected state $s_{next}$, Frontier $\mathcal{F}$, Exploration count $n$
**Ensure:** Trajectory set $\mathcal{T}_s$
1: $\mathcal{T}_s \leftarrow \emptyset$
2: $W_{local} \leftarrow \emptyset$
3: **for** $i = 1$ to $n$ **do**
4: $\quad \tau_i \leftarrow \pi_{explore}(s_{next}, W_{local}, \mathcal{T}_s, \mathcal{F})$
5: $\quad \mathcal{T}_s \leftarrow \mathcal{T}_s \cup \{\tau_i\}$
6: $\quad W_{local} \leftarrow MAR(\mathcal{T}_s, \mathcal{F})$
$\quad$ where $W_{local} = \{(s_1^*, A_{s_1^*}), ..., (s_k^*, A_{s_k^*})\}$
7: **end for**
8: **return** $\mathcal{T}_s$

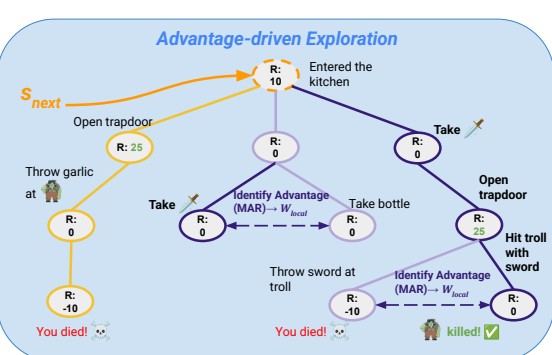

Figure 2: (a) Explore procedure in GLoW, (b) Illustration of exploration with Local World Memory

**Semantic Advantage Representation** Concretely, MAR is an LLM operation taking the local exploration trajectories $\mathcal{T}_s$ and frontier trajectories $\mathcal{F}$ as inputs, and generating the structured textual output $W_{local} = \{(s_1^*, A_{s_1^*}), ..., (s_k^*, A_{s_k^*})\}$, where $s_1^*, ..., s_k^*$ are critical states (typically 2-4) and

---

[2]Note that this assumes a deterministic environment. We discuss this limitation and possible stochastic extensions in Appendix I.

each $A_{s_i^*}$ encodes *semantic advantages*. MAR features two design principles for enhancing the accuracy of semantic advantage inference: First, multi-trajectory comparison enables LLM reasoning to aggregate over divergent outcomes revealing good/bad actions, or consistent patterns confirming reliable strategies, while focusing analysis on critical states where these signals are most informative. Second, the frontier trajectories (representing the best outcomes achieved so far) provide a stable reference point, grounding the LLM's evaluation of whether new trajectories constitute meaningful progress. This implements a functional role analogous to a value baseline through context-based reasoning rather than numerical subtraction.[3]

Unlike scalar advantages $A(s, a)$, these semantic advantages capture progress signals not expressed by sparse rewards, while serving an analogous functional role by guiding exploration policy through $W_{local}$. We provide a full example of $W_{local}$ generated for Zork1 in Appendix F.2.

**Exploration Policy** The local world memory module enhances the exploration phase by guiding a policy defined by an LLM agent, as:

$$\pi_{\text{explore}}(a|s_t, h_t) = \text{Agent}_{\text{LLM}}(s_t, h_t, W_{\text{local}}, T_s, \mathcal{F}) \qquad (3)$$

where $h_t$ is the current trajectory history, $T_s$ contains previous trajectories in the same exploration phase, and the policy leverages both learned advantages from $W_{\text{local}}$ and successful strategies from frontier $\mathcal{F}$. Fig. 2 illustrates exploration in GLoW with the local world memory module. Consider a trajectory (gold) that reached the cellar but failed at the troll bottleneck without the sword. Based on the global world memory (Fig. 1), which identifies high $v'$ at the cellar state, this state becomes $s_{next}$ (orange root, Fig. 2). The local world memory guides multiple exploration attempts (purple paths), where MAR identifies advantages for "taking sword" despite no immediate reward. This advantage learning guides successful exploration through the troll bottleneck (rightmost path). Finally, to address Jericho's exponential action space, we implement a hybrid approach combining free generation with soft constraints given by valid actions from the Jericho environment (further details are provided in Appendix E.4).

## 4 RESULTS

We evaluate GLoW on the Jericho benchmark suite. We next present baselines (Sec. 4.1), experimental setup (Sec. 4.2), main results demonstrating the effectiveness of GLoW (Sec. 4.3), and ablation studies (Sec. 4.4) isolating each module contribution. Lastly, we provide detailed analysis of exploration dynamics in (Sec. 4.5).

### 4.1 BASELINES

We perform comprehensive comparison against baselines spanning RL-based, MCTS-based, and LLM-based approaches. Furthermore, we compare with specialized methods for hard-exploration problems in each type of baseline. All methods assume access to valid actions from Jericho.

**RL-Based Methods: DRRN** (He et al., 2016) is a value-based RL approach for choice-based games, learning Q-values for valid actions using GRU encoders and decoders trained via TD loss. **KG-A2C** (Ammanabrolu & Hausknecht, 2020) is an on-policy RL agent that adapts Advantage Actor Critic (A2C) (Mnih et al., 2016), augmented by a dynamic knowledge graph as a state representation that is learned during exploration. Similar to DRRN, **RC-DQN** (Guo et al., 2020) is a DQN-based agent (Mnih et al., 2015), but leverages object-centric neural reading comprehension architectures (Seo et al., 2017) for computing Q-values from observations. **eXploit-Then-eXplore (XTX)** (Tuyls et al., 2022) is the current state-of-the-art method in Jericho, implementing Go-Explore with imitation learning on promising trajectories for state selection, and DQN with intrinsic curiosity reward for exploration. RL-based methods rely on million-scale interaction data to learn, leveraging parallel environments for training, with the exception of RC-DQN which leverages 100,000 interactions.

---

[3]In Appendix A, we provide theoretical motivation showing how multi-trajectory comparison with a stable reference can reduce variance in numerical advantage estimation. MAR applies these principles through LLM reasoning rather than explicit numerical computation.

**MCTS-Based Methods:** Monte Carlo Tree Search is widely adopted for large sequential decision-making problems (Browne et al., 2012; Silver et al., 2016), which explores effectively by combining random sampling and tree search. **MC-LAVE** (Jang et al., 2021) combines MCTS with language-driven exploration, concentrating search effort on promising actions identified based on value estimates from semantically similar past actions. **MC-DML** (Shi et al., 2025) enhances MCTS by incorporating LLMs as action priors in the PUCT algorithm (Silver et al., 2016), which balances exploration and exploitation during tree search. The LLM is equipped with a cross-trial memory mechanism, allowing it to learn from past experiences such as death in Zork1. Both methods require around 400,000 environment interactions to build comprehensive search trees.

**LLM-Based Methods: ReAct** (Yao et al., 2023) is the widely adopted standard LLM agent approach interleaving reasoning and acting. **Reflexion** (Shinn et al., 2023) is a multi-episode approach building on ReAct, incorporating self-reflection on each episode to guide future episodes. **In-context Reinforcement Learning (ICRL)** (Song et al., 2026) is another multi-episode approach leveraging in-context reinforcement learning, using cumulative history of past trajectories and rewards as context for future episodes. **Intelligent Go-Explore (IGE)** (Lu et al., 2025) implements Go-Explore with LLMs, leveraging LLM-based state selection from a state archive, combined with ReAct-based exploration. As LLM-based baseline methods were not originally applied on Jericho, we re-implement them for Jericho using the action generation approach with valid action soft-constraint described in Sec. 3.2. All LLM-based approaches use 1,000 interactions to balance performance and API cost. We provide details of LLM API usage and cost in Appendix D.

## 4.2 EXPERIMENTAL SETUP

**Implementation Details** Each method is evaluated over 3 runs, reporting mean and standard deviation of maximum achieved scores. ReAct performs 20 independent 50-step episodes. Reflexion performs 20 trials of 50-step episodes, incorporating sliding-window memories from up to 10 previous attempts. Likewise, ICRL includes a sliding window of 10 previous trajectories as in-context examples. IGE and GLoW adaptively alternate between state selection and 50-step exploration episodes within the total 1,000 step budget. We found 50 steps to be sufficient for baseline agents, as they typically plateau early on puzzles or repetitive action loops before reaching this limit. We use temperature 0.5 for all methods except IGE, which uses 0.3 following Lu et al. (2025). For GLoW hyperparameters, $n$=3 exploration trajectories and $k$=5 trajectory frontier size is used.

**Evaluation** We evaluate on 10 games from the Jericho benchmark (Hausknecht et al., 2020), spanning different difficulty levels. Jericho's games feature massive combinatorial state-action spaces (e.g., 110 rooms and 70 interactable objects in Zork1) with no explicit task instructions, requiring agents to discover intermediate goals and hidden causal dependencies through exploration. Furthermore, as an established benchmark suite evaluated by several prior works, it enables direct comparison with RL, MCTS, and LLM-based approaches. Following the benchmark's categorization, we test on *possible games* (Pentari, Detective, Temple, Ztuu) featuring moderate puzzles and frequent rewards, *difficult games* (Zork1, Zork3, Deephome, Ludicorp) requiring more complex inventory management, puzzle-solving and navigation, and *extreme games* (Enchanter) involving non-standard actions and spell mechanics. We use the standard Jericho interface providing textual observations and access to valid actions at each step. Unlike some prior work, we do not augment observations with explicit "look" or "inventory" commands, instead allowing agents to learn these through play.

## 4.3 MAIN RESULTS

We report our main results in Table 1. GLoW achieves a new state-of-the-art performance among LLM approaches across 7 out of 10 games. On Zork1, a canonical game of the Jericho suite, our method reaches a score of 73.0, a significant improvement over the next best LLM method (ICRL at 51.7), and surpassing all compared approaches (with the exception of XTX), including RL and MCTS baselines that use orders of magnitude more interactions. We observe the same strong improvements over the closest LLM method in Ludicorp (73.7 vs. 32.0 for ICRL), Enchanter (61.7 vs. 50.0 for IGE), Ztuu (29.3 vs. 18.7 for ReAct), and Balances (16.7 vs. 11.7 for ICRL).

Notably, our implementation of baselines with hybrid action generation approach shows surprisingly strong performance, whereas prior works reported near-zero scores for LLM agents on Jericho (Shi et al., 2025; Cui et al., 2025; Phan et al., 2025). Our implementation enables ReAct, Reflexion

| Games | RL-based | | | | MCTS-based | | LLM-based | | | | |
|---|---|---|---|---|---|---|---|---|---|---|---|
| | **DRRN** | **KG-A2C** | **RC-DQN** | **XTX** | **MC-LAVE** | **MC-DML** | **ReAct** | **Reflexion** | **ICRL** | **IGE** | **GLoW (Ours)** |
| Steps | 1,000,000 | 1,600,000 | 100,000 | 800,000 | ~400,000 | ~400,000 | 1000 | 1000 | 1000 | 1000 | 1000 |
| Enchanter | 20 | 12.1 | 20 | $\underline{52.0}$ | – | $20_{\pm0.0}$ | $46.7_{\pm9.4}$ | $48.3_{\pm9.4}$ | $43.3_{\pm8.5}$ | $50.0_{\pm7.1}$ | $\mathbf{61.7}_{\pm20.1}$ |
| Zork1 | 32.6 | $40.2_{\pm0.4}$ | 38.8 | $\mathbf{103.4}_{\pm10.9}$ | 45.2 | $48.66_{\pm1.89}$ | $48.3_{\pm4.7}$ | $48.0_{\pm5.0}$ | $51.7_{\pm4.7}$ | $44.3_{\pm0.5}$ | $\underline{73.0}_{\pm4.5}$ |
| Zork3 | 0.5 | 0.0 | 2.83 | $\underline{4.2}_{\pm0.1}$ | – | $3_{\pm0.0}$ | $3.0_{\pm0.0}$ | $2.7_{\pm0.5}$ | $3.0_{\pm0.0}$ | $3.7_{\pm0.9}$ | $\mathbf{4.3}_{\pm0.9}$ |
| Deephome | 1 | $20_{\pm2.1}$ | 1 | $\mathbf{77.7}_{\pm2.1}$ | 35 | $67_{\pm1.41}$ | $11.0_{\pm4.2}$ | $22.0_{\pm1.6}$ | $24.0_{\pm5.7}$ | $71.3_{\pm4.9}$ | $\underline{75.0}_{\pm8.7}$ |
| Ludicorp | 13.8 | $19.8_{\pm1.0}$ | 17 | **78.8** | 22.8 | $19.67_{\pm1.7}$ | $19.7_{\pm0.9}$ | $21.7_{\pm1.2}$ | $32.0_{\pm7.1}$ | $28.3_{\pm11.3}$ | $\underline{73.7}_{\pm11.0}$ |
| Balances | 10 | 10 | 10 | **24** | 10 | $10_{\pm0.0}$ | $10_{\pm0.0}$ | $10_{\pm0.0}$ | $11.7_{\pm2.4}$ | $10.0_{\pm0.0}$ | $\underline{16.7}_{\pm2.4}$ |
| Pentari | 27.2 | $44_{\pm0.9}$ | 43.8 | 49.6 | $\underline{68}$ | $\mathbf{70}_{\pm0.0}$ | $30.0_{\pm0.0}$ | $30.0_{\pm0.0}$ | $26.7_{\pm4.7}$ | $30.0_{\pm0.0}$ | $30.0_{\pm0.0}$ |
| Detective | 197.8 | $\underline{338}_{\pm3.4}$ | 291.3 | 312.2 | 330 | $\mathbf{346.67}_{\pm9.43}$ | $113.3_{\pm4.7}$ | $166.7_{\pm20.5}$ | $233.3_{\pm47.8}$ | $316.7_{\pm4.7}$ | $310.0_{\pm8.2}$ |
| Temple | 7.4 | 8 | 8 | – | $8_{\pm0.0}$ | $8_{\pm0.0}$ | $8.7_{\pm0.9}$ | $8.7_{\pm0.9}$ | $8_{\pm0.0}$ | $\mathbf{13.7}_{\pm0.9}$ | $\underline{13.0}_{\pm0.0}$ |
| Ztuu | 21.6 | $5_{\pm0.0}$ | – | – | 7 | $\underline{23.67}_{\pm1.9}$ | $18.7_{\pm2.4}$ | $18.3_{\pm2.6}$ | $16.7_{\pm4.1}$ | $15.0_{\pm9.1}$ | $\mathbf{29.3}_{\pm4.0}$ |

Table 1: Comparison of RL-based, MCTS-based, and LLM-based methods on Jericho benchmark games. We report mean ± standard deviation over 3 runs following prior works (Tuyls et al., 2022; Shi et al., 2025). **Bold** indicates best overall performance, and underline indicates second-best. Steps shows total environment interactions. The color of game names indicates original game difficulty categories from Hausknecht et al. (2020): extreme, difficult, and possible. GLoW achieves state-of-the-art among LLM-based approaches in 7/10 games, and is overall best among all compared approaches in 3/10, second-best in 5/10.

and ICRL to reach 48.3, 48.0, 51.7 on Zork1, respectively, and similarly on par with RL baselines such as KG-A2C and RC-DQN across the board. While this reveals the sample efficiency of LLM agents, these baselines still fall far short of more advanced exploration methods such as XTX and MC-DML, demonstrating the necessity of effective exploration for LLM agents.

Next we compare GLoW against advanced exploration approaches. First, comparing with IGE which is the most directly comparable to ours as an LLM-based Go-Explore method, GLoW substantially outperforms with better performance on 8 out of 10 games. GLoW also achieves competitive performance with state-of-the-art RL and MCTS methods, XTX and MC-DML. We nearly match the overall state-of-the-art XTX, which uses 800× more interactions, on both Deephome (75.0 vs. 77.7) and Ludicorp (73.7 vs. 78.8), and notably surpass it on Enchanter (61.7 vs. 52.0). It also outperforms MC-DML, which employs extensive MCTS-based exploration around 400× more interactions, on most games including Zork1 (73.0 vs. 48.66), Deephome (75.0 vs. 67.0), and Ludicorp (73.7 vs. 19.67). These results demonstrate that our dual-scale approach combining global world memory for value-based state selection, with local advantage learning for exploration, enables significant performance gains in LLM agents, competitive with sample-intensive RL approaches.

## 4.4 ABLATION STUDY

To validate the contribution of each component of GLoW, we perform systematic ablations and report the results in Table 2.

**Effectiveness of Local World Memory** We first analyze the efficacy of our local world memory by ablating MAR. We replace MAR by Reflexion, which performs the same multi-path exploration but does not leverage our proposed advantage learning, instead performing single-trajectory reflection on the latest trajectory. The results show that the performance drops significantly across most games, demonstrating that MAR's advantage-based formulation more effectively leverages multi-trajectory information than Reflexion, improving exploration under sparse rewards.

**Effectiveness of Global World Memory** Next, we analyze the effectiveness of the global world memory module, which consists of the frontier of high-value trajectories, and the LLM-based value analysis and alignment state selection. We first ablate the global world memory $W_{global}$, leveraging the raw frontier trajectories for state selection. The negative performance impact shows that, using LLM to reason across the frontier trajectories to infer potential value is indeed effective. Next, we ablate the trajectory frontier $\mathcal{F}$ altogether, such that it is not used for state selection or leveraged by the exploration policy. This causes further decrease in performance, confirming the contribution of the trajectory frontier in both phases.

| Ablation Variants | Zork1 | Zork3 | Enchanter | Deephome | Ludicorp | Balances |
|---|---|---|---|---|---|---|
| **GLoW (Full)** | **73.0**$_{\pm4.5}$ | **4.3**$_{\pm0.9}$ | **61.7**$_{\pm20.1}$ | **75.0**$_{\pm8.7}$ | **73.7**$_{\pm11.0}$ | **16.7**$_{\pm2.4}$ |
| ✗ [Local WM] Multi-path Advantage Reflection (MAR) | 70.0$_{\pm13.6}$ | 4.3$_{\pm0.5}$ | 51.7$_{\pm9.4}$ | 56.7$_{\pm21.7}$ | 54.7$_{\pm22.4}$ | 11.7$_{\pm2.4}$ |
| ✗ [Global WM] State selection with $W_{global}$ | 62.0$_{\pm15.6}$ | 4.3$_{\pm0.9}$ | 60.0$_{\pm10.8}$ | 61.3$_{\pm26.0}$ | 63.3$_{\pm14.7}$ | 13.3$_{\pm2.4}$ |
| ✗ [Global WM] Trajectory frontier $\mathcal{F}$ | 61.7$_{\pm1.9}$ | 4.0$_{\pm0.8}$ | 53.3$_{\pm10.3}$ | 57.7$_{\pm23.3}$ | 63.3$_{\pm12.3}$ | 11.7$_{\pm2.4}$ |
| ✗ All above | 51.3$_{\pm5.2}$ | 4.3$_{\pm0.9}$ | 51.7$_{\pm9.4}$ | 56.0$_{\pm21.2}$ | 22.0$_{\pm0.8}$ | 10.0$_{\pm0.0}$ |
| Standard IGE | 44.3$_{\pm0.5}$ | 3.7$_{\pm0.9}$ | 50.0$_{\pm7.1}$ | 71.3$_{\pm4.9}$ | 28.3$_{\pm11.3}$ | 10.0$_{\pm0.0}$ |

Table 2: Ablation study on GLoW components. We evaluate the contribution of: (1) Local world memory through MAR, (2) Global world memory for state selection, (3) trajectory frontier $\mathcal{F}$.

**Synergy of Local World Memory and Global World Memory** Finally, we ablate all the above components together. The resultant model is similar to IGE, with multi-path Reflexion for exploration. The results show that simply adding multi-path reflection does not lead to a clear improvement over IGE, indicating that the overall performance of GLoW comes from the complementary synergy of its components.

## 4.5 ANALYSIS

**Controlling global vs. local focus with $n$ exploration parameter** We study the tradeoff between local learning depth and global exploration coverage by varying $n$, the number of explorations per selected state. Larger $n$ enables MAR to learn from more trajectories, while smaller $n$ increases state selection frequency, helping escape local minima. With budget $B$=1000 and steps $s$=50, minimum state selections is $m = \lfloor B/(s \cdot n) \rfloor - 1$. With $n$=1, MAR is turned off. With $n$>1, MAR analyzes $n$-1 local trajectories plus the global frontier trajectories.

Table 3: Controlling the focus on global (less explorations per state but more frequent state selection) vs local learning (more explorations per state). The results demonstrate $n$=3 explorations from promising states strikes a good balance between the two.

| Explorations per State | Max. Steps per Exploration Phase | Min. State Selection | Zork1 | Zork3 | Enchanter | Deephome | Ludicorp | Balances |
|---|---|---|---|---|---|---|---|---|
| 1 (no MAR) | $50 \times 1$ | 19 | 59.0$_{\pm5.7}$ | 3.7$_{\pm0.9}$ | 58.3$_{\pm9.4}$ | 59.7$_{\pm22.6}$ | 34.0$_{\pm15.6}$ | 13.3$_{\pm4.7}$ |
| 2 (MAR w/ 1) | $50 \times 2$ | 9 | 67.3$_{\pm8.7}$ | 3.7$_{\pm1.2}$ | 55.0$_{\pm7.1}$ | 43.3$_{\pm26.6}$ | 66.0$_{\pm3.7}$ | 11.7$_{\pm2.4}$ |
| **3 (MAR w/ 2)** | $50 \times 3$ | 5 | **73.0**$_{\pm4.5}$ | **4.3**$_{\pm0.9}$ | 61.7$_{\pm20.1}$ | 75.0$_{\pm8.7}$ | **73.7**$_{\pm11.0}$ | **16.7**$_{\pm2.4}$ |
| 4 (MAR w/ 3) | $50 \times 4$ | 4 | 63.0$_{\pm6.5}$ | **4.3**$_{\pm0.9}$ | **66.7**$_{\pm10.3}$ | 73.7$_{\pm4.5}$ | 62.0$_{\pm12.4}$ | 16.7$_{\pm2.4}$ |
| 5 (MAR w/ 4) | $50 \times 5$ | 3 | 59.3$_{\pm13.8}$ | 4.0$_{\pm0.8}$ | 46.7$_{\pm6.2}$ | **76.3**$_{\pm6.8}$ | 53.3$_{\pm7.0}$ | 15.0$_{\pm0.0}$ |

Table 3 shows that extreme values of $n$ generally yield suboptimal performance. When n=1, effectively disabling MAR, performance drops significantly on certain games like Ludicorp (34.0 vs 73.7 with $n$=3). Conversely, Deephome shows consistent improvement with increasing $n$, suggesting it particularly benefits from deeper local exploration. The results demonstrate that moderate increases in $n$ improve performance across several games, consistent with our theoretical motivation (Appendix A) that MAR should benefit from multi-trajectory comparisons. However, setting $n$=5 begins to degrade performance, as excessive commitment to individual exploration phases reduces minimum state selection frequency to just 3, increasing susceptibility to local optima. These findings indicate that balancing global and local learning is crucial. We select $n$=3 as our default parameter, as it achieves the best overall performance by providing sufficient trajectories for robust advantage estimation while maintaining adequate state selection frequency to escape local minima.

## 4.6 SCALING WITH STRONGER LLMS

To validate that GLoW generalizes across model capabilities, we further evaluate all LLM-based methods using GPT-4.1 on the 6 Extreme/Difficult games. Notably, we find that our method establishes a new state-of-the-art across all existing agents on 4 out of 6 games, even surpassing XTX on 5 out of 6 games, while using 800× fewer interactions. These results demonstrate the robustness of our proposed approach across LLM capabilities. We provide further qualitative analysis of how model capability affects $W_{\text{global}}$ quality in Appendix G, and analyze failure modes of GLoW in Appendix H.

| Games | RL | LLM-based | | | | | | | | | |
|-------|-----|-----------|--|--|--|--|--|--|--|--|--|
| | | GPT-4.1 mini | | | | | GPT-4.1 | | | | |
| | XTX | ReAct | Rfl | ICRL | IGE | GLoW | ReAct | Rfl | ICRL | IGE | GLoW |
| Steps | 800K | 1K | 1K | 1K | 1K | 1K | 1K | 1K | 1K | 1K | 1K |
| Enchanter | 52.0 | $46.7_{\pm9.4}$ | $48.3_{\pm9.4}$ | $43.3_{\pm8.5}$ | $50.0_{\pm7.1}$ | $61.7_{\pm20.1}$ | $38.3_{\pm2.4}$ | $58.3_{\pm2.4}$ | $45_{\pm7.1}$ | $\underline{68.3_{\pm18.4}}$ | $\mathbf{98.3_{\pm4.7}}$ |
| Zork1 | $\mathbf{103.4_{\pm10.9}}$ | $48.3_{\pm4.7}$ | $48.0_{\pm5.0}$ | $51.7_{\pm4.7}$ | $44.3_{\pm0.5}$ | $73.0_{\pm4.5}$ | $45.0_{\pm0.0}$ | $54.3_{\pm4.5}$ | $48.0_{\pm2.8}$ | $86.7_{\pm24.1}$ | $\underline{103.0_{\pm6.8}}$ |
| Zork3 | $4.2_{\pm0.1}$ | $3.0_{\pm0.0}$ | $2.7_{\pm0.5}$ | $3.0_{\pm0.0}$ | $3.7_{\pm0.9}$ | $\underline{4.3_{\pm0.9}}$ | $3.3_{\pm0.5}$ | $2.7_{\pm0.5}$ | $3.0_{\pm0.8}$ | $3.0_{\pm0.0}$ | $\mathbf{5.0_{\pm0.0}}$ |
| Deephome | $77.7_{\pm2.1}$ | $11.0_{\pm4.2}$ | $22.0_{\pm1.6}$ | $24.0_{\pm5.7}$ | $71.3_{\pm4.9}$ | $75.0_{\pm8.7}$ | $32.3_{\pm19.6}$ | $22.3_{\pm1.7}$ | $34.7_{\pm18.7}$ | $\underline{82.0_{\pm8.6}}$ | $\mathbf{114.7_{\pm27.8}}$ |
| Ludicorp | 78.8 | $19.7_{\pm0.9}$ | $21.7_{\pm1.2}$ | $32.0_{\pm7.1}$ | $28.3_{\pm11.3}$ | $73.7_{\pm11.0}$ | $31.0_{\pm2.8}$ | $29.0_{\pm0.8}$ | $31.7_{\pm0.5}$ | $\mathbf{89.0_{\pm7.8}}$ | $\underline{79.0_{\pm16.8}}$ |
| Balances | $\underline{24}$ | $10_{\pm0.0}$ | $10_{\pm0.0}$ | $11.7_{\pm2.4}$ | $10.0_{\pm0.0}$ | $16.7_{\pm2.4}$ | $18.3_{\pm2.4}$ | $18.3_{\pm2.4}$ | $16.7_{\pm2.4}$ | $16.7_{\pm2.4}$ | $\mathbf{26.7_{\pm2.4}}$ |

Table 4: Results comparing GPT-4.1 mini and GPT-4.1 on Extreme/Difficult games. We include XTX, the strongest RL baseline, for reference.

## 5 RELATED WORKS

**Go-Explore-based Methods** Go-Explore (Ecoffet et al., 2021) enables effective exploration in sparse-reward environments by decomposing exploration into state selection and exploration phases. IGE (Lu et al., 2025) adapts Go-Explore for LLMs, using LLM-based "promisingness" for state selection and ReAct for exploration. However, IGE's limited exploration and ill-defined selection criteria limit its effectiveness in complex environments like Jericho. Our work addresses these limitations through principled value decomposition for selection, and multi-path advantage learning for exploration.

**Agents for Text-based Games** RL approaches to Jericho include DRRN (He et al., 2016), KG-A2C (Ammanabrolu & Hausknecht, 2020), and RC-DQN (Guo et al., 2020), and the aforementioned XTX, where all are sample-intensive, relying on hundreds of thousands of interactions. MCTS-based methods like MC-LAVE (Jang et al., 2021) and MC-DML (Shi et al., 2025) leverage tree search but still rely on a similar scale of interactions. We show that LLM agents can achieve comparable performance to RL methods, while requiring orders of magnitude fewer interactions through structured exploration and learning mechanisms.

**Learning in LLM Agents** Recent works have studied how LLMs can learn from experience. Reflexion (Shinn et al., 2023) enables learning through self-reflection on failed attempts, while in-context reinforcement learning (ICRL) (Song et al., 2026) leverages previous trajectories' history as context. However, these approaches struggle with sparse rewards due to noisy learning signals. Our MAR mechanism addresses this challenge through multi-path advantage-based learning, providing more robust learning signals.

**World Models for LLM Agents** While traditional world models in model-based RL focus on transition dynamics (Ha & Schmidhuber, 2018; Hafner et al., 2024), recent works adopt an expanded paradigm of world models as mechanisms for implicit representations of task-relevant knowledge (Ding et al., 2025; Li et al., 2024). Li et al. (2024) formalize this notion through state abstraction theory (Abel, 2022), showing that effective LLM agents build goal-oriented abstractions without recovering full dynamics. GLoW 's dual-scale world memory draws from this view, where the global world memory extracts value decompositions across global discoveries, while the local world memory captures semantic advantage signals for exploration.

## 6 CONCLUSION

We introduce GLoW, a dual-scale world memory framework to tackle hard-exploration problems. GLoW leverages a global world memory that enables principled decomposition of state values, and a local world memory that integrates trajectories from the same state as controlled exploration feedbacks. Our approach achieves state-of-the-art performance among LLM methods on the challenging Jericho benchmark, while matching RL-based methods that require 800× more environment interactions. By learning global value patterns across discoveries, and local progress signals from multi-path exploration, GLoW overcomes a key limitation of LLM agents in hard-exploration tasks, demonstrating sample-efficient yet high performance results.

## REPRODUCIBILITY STATEMENT

To ensure reproducibility of our results, we provide comprehensive implementation details in the paper. Algorithm 4 provides the complete pseudocode for GLoW, and hyperparameters are detailed in Section 4.2 ($n$=3 exploration trajectories, temperature=0.5, $k$=5 frontier size, 1000 environment steps). All prompts used for the global world memory (Appendix E.1), LLM-based state selection (Appendix E.2), MAR (Appendix E.3), and exploration policy (Appendix E.4) are provided in full. Experiments use `gpt-4.1-mini-2025-04-14` as the LLM backbone, with additional experiments using `gpt-4.1-2025-04-14` to evaluate scaling behavior, reporting results averaged over 3 runs with standard deviations. We implement all LLM baselines using the same action generation approach (Section 3.2) for fair comparison. The Jericho benchmark is publicly available, and we use the standard evaluation protocol from Hausknecht et al. (2020). Code implementation is publicly released.

## ACKNOWLEDGMENTS

This work was partly supported by Institute of Information & communications Technology Planning & Evaluation (IITP) grant funded by the Korea government(MSIT) [NO.RS-2021-II211343, Artificial Intelligence Graduate School Program (Seoul National University)], and the National Research Foundation of Korea(NRF) grant funded by the Korea government(MSIT) (No. RS-2024-00414981), and the MSIT(Ministry of Science and ICT), Korea, under the ITRC(Information Technology Research Center) support program(IITP-2025-2020-0-01789) supervised by the IITP(Institute for Information & Communications Technology Planning & Evaluation).

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

## A  THEORETICAL MOTIVATION FOR MULTI-PATH ADVANTAGE REFLECTION

### A.1  VARIANCE REDUCTION THROUGH MULTI-TRAJECTORY COMPARISON

The design of MAR draws motivation from classical results on variance reduction in advantage estimation. We present this theoretical background that inspired our approach, noting that MAR implements these principles through LLM reasoning rather than explicit numerical computation.

**Proposition 1.** *Consider $n$ trajectories $\{\tau_1, ..., \tau_n\}$ starting from state $s$. For any state $s^*$ visited by $m \leq n$ of these trajectories, an advantage estimate computed by averaging over $m$ trajectory outcomes has variance reduced by factor $1/m$ compared to a single-trajectory estimate, assuming bounded variance across trajectories:*

$$\text{Var}[\hat{A}_{\text{multi}}(s^*)] \leq \frac{\text{Var}[\hat{A}_{\text{single}}(s^*)]}{m}$$

**Proof.** For a trajectory $j$ passing through state $s^*$ and taking action $a_j$, let $R_j(s^*, a_j)$ denote the random variable representing the sum of future rewards from $s^*$ onward. This provides an unbiased estimate of the true $Q(s^*, a_j)$.

The single-trajectory advantage estimate for action $a$ is:

$$\hat{A}_{\text{single}}(s^*, a) = R_j(s^*, a) - \hat{V}(s^*)$$

where $\hat{V}(s^*)$ is an estimate of the state value. This estimate has high variance because it relies on a single sample: $\text{Var}[\hat{A}_{\text{single}}(s^*, a)] = \text{Var}[R_j(s^*, a)]$ when $\hat{V}(s^*)$ is held constant.

Now consider a multi-trajectory approach. From the $m$ trajectories passing through $s^*$, let $m_a$ denote the number of trajectories taking action $a$. Averaging outcomes yields an improved Q-value estimate:

$$\hat{Q}_{\text{multi}}(s^*, a) = \frac{1}{m_a} \sum_{j:a_j=a} R_j(s^*, a)$$

Using basic properties of variance for independent random variables with equal variance $\sigma_a^2$:

$$\text{Var}[\hat{Q}_{\text{multi}}(s^*, a)] = \text{Var}\left[\frac{1}{m_a} \sum_{j:a_j=a} R_j\right] = \frac{1}{m_a^2} \cdot m_a \cdot \sigma_a^2 = \frac{\sigma_a^2}{m_a}$$

This shows variance reduction by factor $m_a$ for the Q-estimate. For the baseline, incorporating a stable reference (such as outcomes from previously successful trajectories) rather than a fluctuating estimate further reduces variance. Under the assumption that this stable baseline has low variance relative to the Q-estimate, the variance of the advantage estimate is dominated by the Q-component:

$$\text{Var}[\hat{A}_{\text{multi}}(s^*, a)] \approx \text{Var}[\hat{Q}_{\text{multi}}(s^*, a)] = \frac{\sigma_a^2}{m_a} \leq \frac{\sigma_a^2}{1} = \text{Var}[\hat{A}_{\text{single}}(s^*, a)]$$

For any action with $m_a \geq 1$ samples, we achieve variance reduction by a factor of $m_a$. $\square$

### A.2 CONNECTION TO MAR

The proposition above motivates why multi-trajectory comparison with a stable baseline can reduce variance in advantage estimation. While not directly approximating the numerical quantity, MAR operationalizes these principles through LLM reasoning rather than explicit numerical computation:

**Multi-trajectory aggregation.** Rather than computing the average $\frac{1}{m_a} \sum_j R_j$, MAR prompts the LLM to reason across multiple trajectories from the same starting state, identifying consistent patterns and divergent outcomes. This achieves a benefit analogous to variance reduction through averaging, as the LLM can implicitly weigh evidence from multiple outcomes when inferring which actions led to better progress.

**Stable baseline via frontier.** The frontier $\mathcal{F}$ serves an analogous role to the stable $\hat{V}(s^*)$ in the proposition. Like target networks in DQN (Mnih et al., 2015) that update periodically to provide stable targets, $\mathcal{F}$ updates only when superior trajectories are discovered, providing a consistent reference point for the LLM's evaluation of whether new trajectories constitute meaningful progress.

## B ALGORITHMS

We provide the full algorithm of GLoW in Alg. 4.

## C CONTAMINATION CHECK

Table 5: Data contamination analysis: LLM accuracy (%) on navigation questions without seeing gameplay.

| Game | # Questions | Accuracy (%) | | |
| --- | --- | --- | --- | --- |
| | | GPT-4.1-nano | GPT-4.1-mini | GPT-4.1 |
| Zork1 | 230 | 11.7 | 10.9 | 13.5 |
| Zork3 | 194 | 9.8 | 8.2 | 8.8 |
| Enchanter | 239 | 10.0 | 9.2 | 9.2 |
| Detective | 66 | 13.6 | 9.1 | 15.2 |
| Balances | 54 | 5.6 | 1.9 | 1.9 |
| Pentari | 70 | 1.4 | 1.4 | 1.4 |
| Deephome | 288 | 13.5 | 17.0 | 14.9 |
| Temple | 92 | 19.6 | 12.0 | 8.7 |
| Ludicorp | 320 | 22.5 | 19.7 | 20.0 |
| Ztuu | 71 | 8.5 | 9.9 | 11.3 |
| Avg. | | 13.0 | 10.4 | 10.9 |

To assess whether large language models have prior knowledge of Jericho games, we conducted a data contamination analysis following the methodology of Tsai et al. (2025). We evaluate contamination by testing whether models can navigate between locations without being shown any gameplay. Specifically, we: (1) collect a walkthrough trajectory by executing up to 300 steps from each game's built-in Jericho walkthrough actions, (2) build a graph of locations and transitions from this walkthrough, (3) generate navigation questions asking for paths between observed locations, and (4) query the model with these questions without providing any context. Navigation questions

---

**Algorithm 4** GLoW: **G**lobal-**Lo**cal **W**orld Memory

---

1: **procedure** GLoW($s_0, n_{iter}, n, k$)
2:     $\mathcal{F} \leftarrow \emptyset$                                                      ▷ Initialize frontier
3:     $\mathcal{A} \leftarrow \{(s_0, 0)\}$                                        ▷ Initialize state archive
4:     **for** $i = 1$ to $n_{iter}$ **do**
5:         $s_{\text{next}} \leftarrow \text{SELECTSTATE}(\mathcal{F}, \mathcal{A})$
6:         $\mathcal{T} \leftarrow \text{EXPLORE}(s_{\text{next}}, \mathcal{F}, n)$
7:         $\text{UPDATEARCHIVE}(\mathcal{T}, \mathcal{F}, \mathcal{A}, k)$
8:     **end for**
9:     **return** $\arg\max_{\tau \in \mathcal{F}} v(\tau)$
10: **end procedure**
11:
12: **procedure** SELECTSTATE($\mathcal{F}, \mathcal{A}$)
13:     $W_{\text{global}} \leftarrow g_{\text{LLM}}(\mathcal{F})$
14:     $s_{\text{next}} \leftarrow \arg\max_{s \in \mathcal{A}} \text{align}_{\text{LLM}}(s, W_{\text{global}})$       ▷ Select state based on decomposed value
15:     **return** $s_{\text{next}}$
16: **end procedure**
17:
18: **procedure** EXPLORE($s, \mathcal{F}, n$)
19:     $\mathcal{T} \leftarrow \emptyset$                               ▷ Initialize trajectory set for current exploration phase
20:     $W_{\text{local}} \leftarrow \emptyset$
21:     **for** $j = 1$ to $n$ **do**
22:         $\tau_j \leftarrow \pi_{\text{explore}}(s, W_{\text{local}}, \mathcal{T}, \mathcal{F})$              ▷ Rollout full trajectory from $s$
23:         $\mathcal{T} \leftarrow \mathcal{T} \cup \{\tau_j\}$
24:         $W_{\text{local}} \leftarrow \text{MAR}(\mathcal{T}, \mathcal{F})$
25:     **end for**
26:     **return** $\mathcal{T}$
27: **end procedure**
28:
29: **procedure** MAR($\mathcal{T}, \mathcal{F}$)
30:     $W_{\text{local}} \leftarrow f_{\text{LLM}}(\mathcal{T}, \mathcal{F})$               ▷ Extract semantic advantages at key states
31:     **return** $W_{\text{local}}$
32: **end procedure**
33:
34: **procedure** UPDATEARCHIVES($\mathcal{T}, \mathcal{F}, \mathcal{A}, k$)
35:     **for** $\tau \in \mathcal{T}$ **do**
36:         $\mathcal{F} \leftarrow \text{top-}k(\mathcal{F} \cup \{\tau\}, v)$             ▷ Update the trajectory frontier
37:         **for** $s' \in \tau$ **do**
38:             $\mathcal{A} \leftarrow \mathcal{A} \cup \{(s', score(s'))\}$        ▷ Add states to state archive
39:         **end for**
40:     **end for**
41: **end procedure**

---

take the form: "In [GAME], what steps would you take to go to [LOCATION B] from [LOCATION A]?" We evaluate responses using strict pattern matching with word boundaries, requiring the exact sequence of navigation commands to appear consecutively in the model's response.

Table 5 shows results of contamination checks for `GPT-4.1-nano`, `GPT-4.1-mini`, and `GPT-4.1` across 10 Jericho games. We observe minimal contamination across all three models, with all individual game accuracies remaining below 23%. The majority of games show less than 15% accuracy across all models, consistent with random guessing or generic text adventure knowledge. The slightly higher accuracies for Ludicorp (up to 22.5%) and Deephome (up to 17.0%) likely reflect the model providing common navigation commands (e.g., "go south") that occasionally match by chance. Notably, scaling model capacity from nano to the full GPT-4.1 does not meaningfully increase contamination, suggesting that these results reflect generic text adventure heuristics rather than memorized game knowledge. Even famous games like Zork1 show accuracy near chance level (10.9–13.5%), while less-known games like Balances (1.9–5.6%) and Pentari (1.4%) show essentially no prior knowledge. These low accuracy rates, combined with the model's generic responses that lack game-specific details, indicate that our experimental results reflect genuine exploration and reasoning capabilities rather than memorized solutions.

# D LLM API COST

We use `gpt-4.1-mini-2025-04-14` for all LLM components ($0.40/$1.60 per million input/output tokens). Per-run costs of all LLM-based approaches with 1,000 environment steps range from $1 to $7, maintaining practicality for research iteration. For experiments with stronger LLMs, we use `gpt-4.1-2025-04-14` for all LLM components ($2.00/$8.00 per million input/output tokens). Per-run costs range from $7.5 to $45.00, demonstrating that while stronger models increase costs by 5-6×, the relative efficiency of our approach remains consistent with ∼40% fewer tokens than ICRL, while achieving superior performance.

Table 6: Comparison of LLM API costs with GPT-4.1-mini. We report the average token consumption and costs across 6 games (Zork1, Zork3, Deephome, Ludicorp, Detective, Temple).

| Method | Input tokens | Output tokens | Total tokens | Avg. cost/run |
|---|---|---|---|---|
| ICRL | 17.6M | 43.1K | 17.6M | $7.10 |
| **GLoW** | 10.9M | 115.1K | 11.0M | $4.54 |
| Reflexion | 7.2M | 50.6K | 7.3M | $2.98 |
| IGE | 3.9M | 44.1K | 3.9M | $1.61 |
| ReAct | 2.6M | 35.5K | 2.6M | $1.08 |

Table 7: Comparison of LLM API costs with GPT-4.1. We report the average token consumption and costs across the same 6 games.

| Method | Input tokens | Output tokens | Total tokens | Avg. cost/run |
|---|---|---|---|---|
| ICRL | 22.4M | 55.1K | 22.5M | $45.24 |
| **GLoW** | 13.3M | 126.2K | 13.4M | $27.58 |
| Reflexion | 9.9M | 62.5K | 10.0M | $20.31 |
| IGE | 4.9M | 64.2K | 5.0M | $10.33 |
| ReAct | 3.5M | 58.2K | 3.5M | $7.45 |

# E PROMPTS

We present the full prompts used in GLoW. Our prompts rely solely on simple instructions and structured output formats without requiring few-shot exemplars, enabling the method to generalize across diverse game scenarios.

## E.1 FRONTIER TRAJECTORY ANALYSIS

**Analysis ($g_{\text{LLM}}$) Prompt**

```
Analyze these successful game trajectories to identify patterns and
strategy:

{For each trajectory in F:}
Trajectory N (Peak:  X, Final:  Y):
  [score] action -> observation (reward:  +/-N if non-zero)
  [score] action -> observation
  ...

Based on these trajectories, provide a strategic analysis:

1.  FRONTIER & EXPLORATION STATUS:
  - What areas/locations have been successfully reached?
  - What remains unexplored or inaccessible?

2.  GAME CHECKPOINTS & PROGRESS:
```

```
   - What are the key milestones/checkpoints identified?
   - What items or abilities unlock new areas?
   - What phase of the game are we in?

3.  BOTTLENECKS & CHALLENGES:
   - Where do trajectories commonly get stuck?
   - What obstacles block further progress?
   - What resources or knowledge are we missing?

4.  REWARD STRUCTURE:
   - When and how are points earned?
   - What actions yield the highest rewards?
   - Are there patterns to the scoring?

5.  NEXT INVESTIGATION GOALS:
   - What specific objectives should we pursue?
   - Which unexplored areas are most promising?
   - What items or states do we need to reach?

Provide a concise strategic summary focusing on actionable
insights.
```

## E.2 STATE SELECTION

**State Selection $(\text{align}_{\text{LLM}})$ Prompt**

```
=== STRATEGIC GAME ANALYSIS ===
{Analysis of frontier trajectories $W_{\text{global}}$}
===================================================

Based on the above analysis, select the state from the archive
that:
- Best aligns with the identified investigation goals
- Can help overcome identified bottlenecks
- Explores promising frontiers
- Has potential for high rewards based on patterns

Current state archive:

0:  [Score:  X, Steps:  Y, Visits:  Z]
  Observation:  {state observation}
  Inventory:  {state inventory}

1:  [Score:  X, Steps:  Y, Visits:  Z]
  Observation:  {state observation}
  Inventory:  {state inventory}

...

Choose state index (0-N).
Respond in JSON format:
{
  "thought":  "Your reasoning about which state best aligns with the
strategic goals",
  "index":  <number>
}
```

## E.3 MULTI-PATH ADVANTAGE REFLECTION (MAR)

The MAR prompt generates $W_{\text{local}}$ as described in Section 3.2, identifying critical decision points and their associated advantages from multiple exploration trajectories. The prompt incorporates

three inputs: (1) the global trajectory frontier containing highest-value trajectories that serve as value baselines, (2) local exploration attempts from the current phase showing different outcomes from the same starting state, and (3) previous $W_{\text{local}}$ outputs when available, enabling cumulative learning within the exploration phase.

By comparing outcomes across these trajectory sources, MAR produces $W_{\text{local}} = \{(s_i^*, A_{s_i^*})\}_{i=1}^k$, identifying where specific actions provide clear advantages. This semantic representation captures causal relationships (e.g., "taking the lamp enables combat in darkness") rather than strictly scalar values, enabling the exploration policy to leverage both statistical patterns from trajectory comparison and LLM reasoning about game mechanics at critical states.

---

**$W_{local}$ Generation Prompt (MAR)**

```
Review these exploration attempts and identify KEY STATE
ADVANTAGES:

{Previous W_local from earlier iterations, if any}

{Global frontier trajectories F}

{Local exploration trajectories from state s}

Analyze all trajectories and identify ADVANTAGES at KEY STATES:

For each important location/state observed across ALL attempts,
list:
- STATE: [description of state/location]
- ADVANTAGES discovered:
  • [specific action] → [specific benefit/outcome] (score impact if
clear)
  • [what to avoid] → [consequence] (score impact if clear)
  • [optimal sequence] → [why it's better]

Example format:
STATE: At the house entrance with lamp
- ADVANTAGES:
  • "go east" → finds sword (enabled +15 points later)
  • "open mailbox first" → gets crucial map (+5 immediate)
  • avoid "go upstairs" early → wastes moves in empty attic (-7
overall)

Focus on:
1.  States that appear across multiple attempts (to see different
outcomes)
2.  Critical decision points where scores diverged significantly
3.  Action sequences that consistently led to success or failure
4.  Items or information that enabled later progress

Provide 2-4 KEY STATES with their discovered advantages.
Be specific about actions, items, and locations from the actual
game.
```

---

### E.4 EXPLORATION POLICY IN GLoW

The prompt for the exploration policy in GLoW is shown below. We instruct the LLM to use a JSON output format with reasoning and action fields.

**Hybrid Action Space** While previous works use either constrained selection from valid actions in RL agents (Hausknecht et al., 2020; Ammanabrolu & Hausknecht, 2020; Tuyls et al., 2022) or pure free-form generation in LLM agents such as ReAct, we provide the valid actions to the LLM as a soft constraint, while still allowing free-form generation. This avoids failure modes of both approaches, where constrained selection can harm action diversity, while pure generation can produce many

invalid actions. As shown in Section 4.2, this hybrid approach, which we use consistently across both GLoW as well as all LLM baselines, significantly improves the base LLM performance with only a lightweight prompt and no few-shot examples.

---

**System Prompt**

```
You are exploring a text adventure game.  Your goal is to make
progress and increase your score.

Generate actions that explore new possibilities and make progress.

Respond in JSON format:
{
  "thought":  "Your reasoning about what to try",
  "action":  "the exact command to execute"
}
```

---

**User Prompt (at initial step)**

```
{Context includes:  W_Local, global frontier trajectories F, previous
exploration trajectories, and the current trajectory.}
================================================

Step 1:
Observation:  {current observation}
Score:  {current score}
Valid actions:  {comma-separated list of valid actions}

What is your next move?
```

---

**User Prompt (subsequent steps)**

```
Step {N}:
Observation:  {current observation}
Score:  {current score}
Valid actions:  {comma-separated list of valid actions}

What is your next move?
```

---

## F  QUALITATIVE EXAMPLES

### F.1  GLOBAL WORLD MEMORY

---

**Global World Memory Example (Zork1)**

**Strategic Analysis of Game Trajectories**

```
================================================
```

**1.  FRONTIER & EXPLORATION STATUS**

**Successfully Reached Areas:**
- Starting point:  Mailbox and House exterior (north, south, east, west of house)
- Forest Path and Clearing with grating
- Up a Tree (bird's nest with egg and leaflet)
- Behind House (window entry)
- Kitchen and Living Room inside the house
- Attic (Trajectory 1)

---

```
- Cellar and Troll Room (Trajectory 1 & 2)

Unexplored or Inaccessible Areas:
- Upstairs beyond the dark staircase (pitch black, dangerous)
- West door with gothic lettering (nailed shut)
- Crawlway south in cellar and forbidding hole west in Troll Room
- Steep metal ramp in cellar (unclimbable)
- Chasm area (Trajectory 1 ends in death)
- Any further passages beyond troll or trap door

==================================================

2.  GAME CHECKPOINTS & PROGRESS

Key Milestones:
- Obtaining jewel-encrusted egg and leaflet from mailbox/tree
- Using egg and leaflet in grating to unlock cellar access
- Entering house via window and acquiring items (food, garlic,
water, lantern, sword, rope, knife)
- Opening trap door and descending into cellar
- Defeating or engaging troll in cellar

Items/Abilities Unlocking New Areas:
- Egg + Leaflet → Grating → Cellar entrance
- Lantern (turned on) → Safe navigation in dark cellar
- Sword (elvish, glowing) → Combat with troll
- Rope and knife (potential utility, not fully leveraged yet)
- Food, garlic, water (likely survival or puzzle items)

Current Phase:  Early-to-mid game - exploration of house and
cellar, initial combat encounters

==================================================

3.  BOTTLENECKS & CHALLENGES

Common Stuck Points:
- Combat with troll (Trajectories 1 & 2 end in death or near death)
- Navigating dark areas without proper light (pitch black upstairs)
- Attempting risky actions (jumping over chasm leads to death)
- Locked or barred doors (gothic door, trap door closing behind
player)

Obstacles Blocking Progress:
- Physical barriers (nailed door, barred trap door)
- Hostile NPC (troll)
- Environmental hazards (darkness, chasm)
- Limited inventory or missing key items

Missing Resources/Knowledge:
- Effective combat strategy or stronger weaponry to defeat troll
safely
- Means to reopen or bypass barred trap door
- Safe traversal methods for chasm or dark upstairs
- Possible puzzle solutions involving rope, knife, or other items

==================================================

4.  REWARD STRUCTURE

When Points Are Earned:
- +5 for taking the egg
- +10 for acquiring food and water items in kitchen
```

```
- +25 for entering cellar (significant milestone)
- +5 for moving north from troll room to passage
- -10 on death and respawn (penalty)

Highest Reward Actions:
- Descending into cellar (+25)
- Collecting key items early (+5 to +10)
- Progressing past major checkpoints

Scoring Patterns:
- Early exploration yields small incremental rewards
- Major area transitions yield large rewards
- Death penalizes score, encouraging cautious play

==================================================

5.  NEXT INVESTIGATION GOALS

Specific Objectives:
- Develop reliable strategy to defeat or bypass troll without dying
- Investigate use of rope and knife for puzzles or combat
- Explore crawlway south and forbidding hole west in troll room
- Find safe method to cross or bypass chasm
- Attempt to open or circumvent nailed gothic door

Most Promising Unexplored Areas:
- Crawlway south and forbidding hole west in cellar/troll room
- Upstairs dark staircase (with lantern or other light source)
- Beyond barred trap door if it can be reopened
- Areas beyond chasm once safe crossing method found
```

## F.2   LOCAL WORLD MEMORY

### Local World Memory Example (Zork1)

```
Based on detailed exploration attempts, here are KEY STATES with
discovered advantages:

==================================================

STATE: The Troll Room
Small room with passages east/south, hole west; troll blocks all
exits; sword glows

ADVANTAGES:
- "hit troll with sword" → kills troll, clears all exits for
exploration (score:  40)
- "take axe" after troll death → obtains valuable weapon for later
puzzles
- avoid engaging without glowing sword → likely failure
- optimal:  kill troll first, then take axe (mandatory to proceed)

==================================================

STATE: Maintenance Room (Flood Control Dam #3)
Room with colored buttons, tool chests, wrench, screwdriver,
toothpaste, leaking pipe

ADVANTAGES:
- "take wrench, screwdriver, tube" → essential tools for
environment interaction
```

```
- "push blue button" → triggers leak, raises water level to access
new areas
- "push red button" → toggles lights, affects water level
- avoid throwing lantern → breaks critical light source
- optimal:  collect tools → manage buttons → control water without
drowning

==================================================

STATE: Temple / Torch Room / Dome Room / Altar
Large temple with inscriptions; dome with railing; rope for
descent; ivory torch; brass bell; gold coffin

ADVANTAGES:
- "take ivory torch" → stable light for deeper cave exploration
- "take bell" → key item for spirit/wraith interaction
- "ring bell at Entrance to Hades" → paralyzes wraiths, enables
passage
- "blow out candles" → enables safe descent or passage
- optimal:  acquire torch → bell → sceptre → manipulate altar →
control spirits

==================================================

STATE: East-West Passage / Chasm Area
Narrow passage with stairs; chasm with paths; multiple routes
(north/east/west/up/down)

ADVANTAGES:
- "east" then "north" → leads to Reservoir South and further areas
- "tie rope to railing" → enables safe descent into lower levels
- avoid getting stuck in loops → wastes moves
- optimal:  explore chasm edges → use rope for vertical → access
Dome/Torch

==================================================

Cross-Cutting Insights:
- Inventory Management:  Strategic dropping/picking essential for
critical artifacts
- Light Preservation:  Maintaining lantern/torch crucial for dark
exploration
- Combat Readiness:  Glowing sword indicates combat opportunity
(essential for progress)
```

## G  Impact of Model Capability on World Memory Quality

To assess the impact of model capability on global world memory quality and trajectory evaluation, we compare three capability scales (GPT-4.1-nano, GPT-4.1-mini, and GPT-4.1) on two games (Zork1 and Deephome). To isolate the effect of model capability, we use an identical frontier trajectory set extracted from a GPT-4.1-mini checkpoint, generating a new $W_{\text{global}}$ with each LLM.

The results are shown below. We present abbreviated trajectory contexts (the inputs for $W_{\text{global}}$ generation), then summarize each $W_{\text{global}}$ into content capturing achieved value ($v$) and potential value ($v'$).

We first observe that the evaluation of $v$ is consistent across model capabilities, indicating that even a smaller model (GPT-4.1-nano) can successfully evaluate trajectories to identify achieved goals and checkpoints. Case Study 1 in Zork1 (*early game, troll puzzle*) serves as the control case, where all three LLMs largely align in both $v$ and $v'$.

In contrast, in other cases we observe clear qualitative differences in the generated $v'$ between GPT-4.1-nano and the stronger models. In Case Study 2 (Zork1 *mid-game*), the trajectories present multiple simultaneous puzzles, and we find that the specificity of reasoning about these objectives shows qualitative differences correlating with model capability. GPT-4.1 and GPT-4.1-mini target specific objectives, while nano generates generic targets. In Case Study 3 (Deephome *early game*), models must infer that Ember (a town mentioned in a note) is the location containing necessary items to proceed. This information must be recalled from reading a note that provides this locational clue. While the larger models successfully make this inference, GPT-4.1-nano fails, resulting in a generic "correct parts for elevators (mentioned in note)," missing the crucial Ember reference.

Since $W_{\text{global}}$ directly influences subsequent state selection, LLM capability gaps in global world memory generation propagate to exploration performance. The average performance across both games reflects these qualitative differences, as shown in Table 8.

Table 8: Impact of model capability on GLoW performance. Scores are mean $\pm$ standard deviation over 3 runs.

| Model | Zork1 | Deephome |
|---|---|---|
| GPT-4.1-nano | $43.0 \pm 2.2$ | $18.3 \pm 5.3$ |
| GPT-4.1-mini | $73.0 \pm 4.5$ | $75.0 \pm 8.7$ |
| GPT-4.1 | $103.0 \pm 6.8$ | $114.7 \pm 27.8$ |

**Case Study 1: Zork1 Early Game**

**Trajectory Context:**
- Trajectories 1, 2 (score 40): house $\rightarrow$ cellar $\rightarrow$ defeat troll $\rightarrow$ maze
- Trajectory 0 (score 15): house exploration, no troll

**Achieved value ($v$):** All models consistently identify *"Defeating the Troll"* as the key milestone.

**Potential value ($v'$):**
- **GPT-4.1-nano:** *"Explore maze, find keys, unlock grating"*
- **GPT-4.1-mini:** *"Navigate maze, find safe chasm crossing, unlock grating"*

**Case Study 2: Zork1 Mid Game**

**Trajectory Context:**
- Trajectories 22, 23 (score 70): dam controls $\rightarrow$ reservoir drains $\rightarrow$ take trunk $\rightarrow$ +15 pts
- Trajectories 15, 16, 18 (score 55): dam area, Loud Room (can't take platinum bar), no trunk
- **Key observations:** Dam puzzles, Loud Room/platinum bar, reservoir access

**Achieved value ($v$):** All models consistently identify key milestones: *"dam controls,"* *"reservoir access,"* and *"trunk acquisition."*

**Potential value ($v'$):**
- **GPT-4.1-nano:** *"Secure tools (wrench, screwdriver, key)"*, *"Explore blocked passages"*, *"Use collected artifacts"* (generic objectives)
- **GPT-4.1-mini:** *"Find how to turn valve at Dam Base"*, *"Explore grating under leaves in clearing"*, *"Explore reservoir as water levels change"* (specific locations and puzzles)
- **GPT-4.1:** *"Solve Loud Room/Platinum Bar puzzle"*, *"Cross chasm to dome room"*, *"Unlock trophy case"* (specific puzzles and objectives)

---

**Case Study 3: Deephome Early Game**

**Trajectory Context:**
- Trajectory 2 (score 20): take note → examine warning → explores to City Gates
- Trajectory 1 (score 8): take note → examine warning → say manaz → death
- Trajectory 0 (score 1): say manaz → immediate death (never reads note)
- **Key observation:** Note text reads *"I traveled south to a human town called **Ember** to procure the correct parts."*

**Achieved value ($v$):** All models consistently identify key milestones: *"reaching City Gates," "discovering control panel,"* and *"reading the note."*

**Potential value ($v'$):**
- **GPT-4.1-nano:** *"correct parts for elevators (mentioned in note)"* (no destination specified)
- **GPT-4.1-mini / GPT-4.1:** *"Obtain elevator parts from **Ember** (human town)"* (specific target location)

---

# H  QUALITATIVE ANALYSIS OF FAILURE MODES

We present a qualitative analysis identifying two primary failure modes observed in GLoW trajectories (GPT-4.1-mini).

**Conservative State Selection:** In Zork1, we observe that GLoW can repeatedly select early-game states (e.g., *State 37: Kitchen, score 15, empty inventory*) despite frontier states with significantly higher scores (e.g., *State 58: Chasm, score 45, with lantern and sword in inventory*) being available. Analysis reveals the model over-prioritizes the acquisition of safety-associated items (*lantern, sword*) from the environment, failing to recognize that frontier states already possess these items in inventory. We posit that this stems from two interacting factors: (a) a logical reasoning failure to verify item possession in the frontier states, and (b) an overvaluation of safety-related semantics, which distorts the estimation of potential value ($v'$ in Eq. 2). This biased selection creates a loop where the agent re-solves completed objectives rather than progressing from the frontier.

**Multi-step Dependency Reasoning Failure:** In Deephome, we illustrate a case where exploration stalls despite unlocking a critical mechanism, due to a failure to resolve subsequent multi-step dependencies. After activating the *City Generator* (step 827, +30 points), which powers the *railway* system, the agent fails to pursue the unlocked paths. The solution requires using the powered *railway* to access intermediate areas (*Blacksmith, Waterfall*) containing prerequisites for further objectives *Water Works* (+30) and *City Gates* (+40). However, the agent attempts the *Water Works* objective directly (2 failed attempts) and reverts to exploring familiar areas, never testing the *railway* (0 visits in the remainder of the game). This reveals a difficulty in systematically resolving multi-step dependencies, particularly when many intermediate objectives are present. We present the two cases in detail below.

---

**Case 1: Zork1**

**Game Context:**
- Reached deeper frontier states through Troll Room to Maze (score 50) and Chasm (score 45).
- State archive includes:
  - State 72: Score 50, Location: Maze, Inventory: lantern, sword, ...
  - State 58: Score 45, Location: Chasm, Inventory: lantern, sword
  - State 37: Score 15, Location: Kitchen, Inventory: empty

**Observed Behavior:**
*Repetitive state selection:*
- Selection Iter. 2: **State 37** (score 15)
- Selection Iter. 3: **State 37** (score 15)
- Selection Iter. 4: **State 37** (score 15)

**Reasoning for state selection:** *"State 37 is in the living room where the brass lantern and the elvish sword are present. Acquiring and activating these items is crucial for safe exploration... This state aligns well with investigation goals..."*

---

---

**Case 2: Deephome**

**Game Context:**
- Agent has activated the City Generator (+30 points), which powers the Railway Station controls.
- Further progress requires using the Railway to access intermediate areas (Blacksmith, Waterfall).

**Observed Behavior:**
- Step 827: Activates generator (+30 points, score 65)
*Failed subsequent multi-stage progression:*
- Steps 828–1000:
  – Railway Station visits: **0**
  – Blacksmith visits: **0**
  – Waterfall visits: **0**
  – Water Works: 2 attempts at wheel, both fail

---

# I ASSUMPTION OF ENVIRONMENT DETERMINISM

Our implementation assumes deterministic environments, which is leveraged by the state restoration mechanism which replays actions to return to a selected state. This limits applicability in stochastic environments where action replay may not return to the intended state. Our trajectory-based design facilitates intuitive extensions to such settings, where a potential approach is framing state restoration as a goal-reaching problem guided by replay trajectories.

We note that the MAR component naturally handles stochasticity since it samples multiple exploration paths from a selected state and performs inference over the paths as a set, enabling the LLM-based analysis to reflect on observed stochastic variations.

# J LLM USAGE

We utilized Claude for minor grammar and language edits in paper writing.

