# OpenReview forum: "Dual-Scale World Memory for LLM Agents towards Hard-Exploration Problems"
_ICLR.cc/2026/Conference — ICLR 2026 Poster_

### Official Review · Reviewer_PRB8 · 2025-10-30

**Soundness:** 2
**Presentation:** 2
**Contribution:** 2
**Rating:** 6
**Confidence:** 3

**Summary:**

This paper introduces GLoW (Global-Local World Models), a new framework for Large Language Model (LLM) agents designed to tackle hard-exploration problems. The core idea is a dual-scale mechanism. At the global scale, GLoW maintains a "trajectory frontier" of the most successful past episodes. An LLM analyzes this frontier to decompose state values into achieved rewards and future potential, guiding the selection of promising states to explore from (similar to the "select" phase of Go-Explore). At the local scale, it employs a "Multi-path Advantage Reflection" (MAR) mechanism, where the agent makes several exploration attempts from a single state. An LLM then compares these attempts to infer "semantic advantages" for specific actions, guiding the subsequent exploration policy (the "explore" phase).

**Strengths:**

The paper's central concept of a dual-scale world model is interesting. At the global scale, it improve the Go-Explore simple state archive with a value-ranked trajectory frontier. This allows an LLM to reason across successful episodes to identify bottlenecks and infer a state's "potential value," providing a more principled method for balancing exploration and exploitation than the vague heuristics of prior work. The local-scale Multi-path Advantage Reflection (MAR) uses the LLM to compare divergent outcomes from multiple exploratory paths starting from the same state. This generates dense, semantic advantage signals, creating an interesting solution for credit assignment in sparse-reward environments and a clear improvement over single-trajectory reflection methods.

**Weaknesses:**

In the established literature of model-based reinforcement learning and robotics, a world model is understood to be a learned, predictive model of an environment's dynamics, specifically its transition function p(s' | s, a) and reward function r(s, a) (Ha & Schmidhuber (2018)). The "world models" presented in this paper function as structured, retrospective representations of experiential knowledge: the global model is a curated repository of successful trajectories, and the local model is a synthesized summary of inferred action advantages. The paper's clarity and contribution would be significantly strengthened by either explicitly justifying this terminological choice or, preferably, by adopting a different terminology.

The experimental setup provides a clear comparison by standardizing on a single LLM (GPT-4.1-mini). While we recognize the practical and budgetary reasons for this choice, a targeted ablation study, even on a subset of the Jericho games with a different LLM, would greatly enhance the paper's contribution. This would provide valuable evidence that the dual-scale mechanism is a robust and generalizable principle, rather than one that relies on the specific idiosyncrasies of the chosen model.

**Questions:**

Could the authors first clarify their choice to use the term 'world model', given that the mechanism is not a predictive dynamics model, as is standard in the RL literature, and this term was notably absent in its predecessor, Go-Explore?

The MAR method is interesting but I guess it would fail in a stochastic setting, do you have any intuition if there is a way to extend it to the stochastic setting?

Have the authors considered the potential of adapting the Multi-path Advantage Reflection mechanism as a powerful reward shaping or exploration module for traditional, sample-based RL agents?

---

> ### Author Response · Authors · 2025-11-23
> **Response to Reviewer PRB8**
>
> Dear reviewer PRB8,
>
> Thank you very much for your insightful review and comments. We are glad to hear that you found our dual-scale world model interesting, with principled state selection and semantic advantage-guided exploration. We appreciate your insightful questions and suggestions, and provide detailed responses to each below:
>
> ---
>
> ### [w1, q1] Justification of World Model terminology, and relation to Go-Explore
>
> Thank you for this important question. Please find our detailed discussion in Common Response 2. While our method does not learn a predictive dynamics model $P(s'|s,a)$, it aligns with the "implicit representation" paradigm of world model, supported by recent literature on goal-oriented state abstractions in LLMs. We respectfully argue that GLoW’s structured abstractions at dual scales fulfill the functional role of a world model in decision-making without full dynamics simulation.
>
> In relation to Go-Explore, these world models are what enables both GLoW’s more principled state selection, and effective exploration, improving over the original algorithm’s heuristic state selection and naive exploration.
>
> ---
>
> ### [w2] Generalizability of GLoW across LLMs
>
> As detailed in our Common Response 1, following your suggestion we conducted additional experiments using GPT-4.1, focusing on the subset of 6 games in the difficult and extreme subsets. We found that GLoW effectively scales with the stronger GPT-4.1, establishing a new state-of-the-art across all existing agents on 4 out of 6 games. On 5 out of 6 games, GLoW surpassed XTX, which leverages 800x more interaction steps. These results validate that our dual-scale mechanism is a robust principle that benefits from stronger reasoning, independent of specific model idiosyncrasies. Thank you again for this suggestion, which further strengthened our evaluation.
>
> ---
>
> ### [q2] Extending MAR to stochastic environments
>
> We appreciate your insightful question. We clarify that the MAR component itself naturally handles stochasticity: Since it samples multiple exploration paths from a selected state and performs inference over the paths as a set, the LLM-based analysis process will be able to reflect upon the observed stochastic variations.
>
> On the other hand, the mechanism for selecting and restoring the starting state for MAR would need modification in stochastic environments. Here, our trajectory-based design facilitates intuitive extensions to handle stochasticity: A reasonable approach would be to treat state restoration as a goal-reaching problem, and design an LLM agent to return to the target state, guided by the replay trajectory as well as additional relevant trajectories as context. We believe such a general solution to the return-to-state problem is a promising direction for future work.
>
> ---
>
> ### [q3] Potential of MAR for intrinsic reward shaping in RL
>
> Thank you for this interesting question. We believe a potential direction is leveraging MAR-inferred advantages to shape intrinsic reward for exploration. Existing approaches in language-based intrinsic reward include: [1] which uses LM-based semantic novelty, [2] which leverages language-based auxiliary question-answering [2], and [3] using a simple but effective LLM annotation of preference between game event captions.
>
> MAR could enhance beyond these methods through its multi-path comparison mechanism. Unlike static preference or novelty, MAR infers advantages by analyzing divergent outcomes from the same state to identify actions that drive progress. This principle could be leveraged to design dense causally-grounded intrinsic rewards that actively guide local exploration in sparse-reward settings.
>
> **References**
>
> [1] Improving Intrinsic Exploration with Language Abstractions. Mu et al., NeurIPS 2022.
>
> [2] EAGER: Asking and Answering Questions for Automatic Reward Shaping in Language-guided RL. Carta et al., NeurIPS 2022.
>
> [3] Motif: Intrinsic Motivation from Artificial Intelligence Feedback. Kilssarov et al., ICLR 2024.

---

### Official Review · Reviewer_pUbB · 2025-10-31

**Soundness:** 3
**Presentation:** 2
**Contribution:** 2
**Rating:** 4
**Confidence:** 3

**Summary:**

This paper addresses the limitation of LLM agents in "hard-exploration" tasks, which are characterized by sparse rewards, deceptive local optima and large state spaces. The authors propose GLOW, a novel framework that learns at two scales to solve this. GLOW addresses the global learning challenge by maintaining a "trajectory frontier" of high-value discoveries, allowing an LLM to analyze these trajectories and select the most promising states for future exploration. To tackle local trial-and-error, its Local World Model uses a Multi-path Advantage Reflection (MAR) mechanism. MAR explores multiple paths from the state selected by the global world model and uses an LLM to infer advantage-like signals. On the Jericho benchmark, GLOW achieves SOTA performance among LLM agents and shows performance comparable to SOTA RL-based methods, with 100-800x fewer environment interactions.

**Strengths:**

- Clear and specific motivation
- Novel framework, combining in-context learning with established RL theories.
- Much higher sample efficiency without compromising performance, a critical advantage in environments where interaction is costly or slow.
- Effective use of LLM reasoning, leveraging LLMs for high-level reasoning to provide dense, semantic feedback from sparse rewards.

**Weaknesses:**

- Heavy reliance on the LLM's reasoning abilities, especially the inferred "semantic advantages" are not as rigorous as traditional, mathematically-derived advantage functions.
- The analysis of efficiency is one-sided, focusing on sample efficiency (environment steps) while obscuring the total LLM calls.
- The choice of experimental environment is not well justified.

#### Minor
- The format of reference is inconsistent.

**Questions:**

- How does the framework's performance differ when using LLMs with different capabilities?
- Can the authors provide a qualitative analysis of failure cases?
- Can the authors justify the reason why the experimental environments (e.g., Game of 24, Put Next To (PN) from BabyAI-Text, and The Cooking Game (CG) from TextWorld), which are used in the closely related work IGE, are not used in this work?
- Can the authors discuss the potential of the proposed method for LLM reasoning? As there are some existing works that also propose different methods to enhance the exploration ability of RLVR algorithms.

I am willing to update my evaluation if my questions and concerns are well addressed during the discussion stage.

---

> ### Author Response · Authors · 2025-11-23
> **Response to Reviewer pUbB**
>
> Dear reviewer pUbB,
>
> Thank you very much for your detailed review and constructive comments. We are glad that you found our framework clearly motivated and novel, combining LLM reasoning with RL theories. We appreciate your insightful questions and comments, and provide detailed responses to your questions and concerns below:
>
> ---
>
> ### [q1] Framework performance with different LLMs
>
> Thank you for this question. As detailed in Common Response 1, we conducted additional experiments evaluating our method with GPT-4.1, and found that GLoW’s effectiveness scales with model capability. Notably, it achieved a new state-of-the-art across all compared methods, surpassing even XTX on 5 of the 6 games tested. These results show that our proposed approach is robust and generalizable.
>
> ---
>
> ### [w1] Rigor and validity of semantic advantages
>
> We acknowledge that a trade-off exists between mathematical rigor and practical effectiveness in LLM-derived semantic advantages. However, we emphasize that semantic advantages are designed specifically to address sparse rewards, where numerical estimation of advantage $A(s,a) = Q(s,a) - V(s)$ is difficult due to high-variance and delayed signals.
>
> Our work draws on two established lines of research. First, recent work on process reward models [1] and RLVR [2] supports our view that dense, advantage-based signals are effective for credit assignment in reasoning tasks where rewards are sparse. Second, research in LLM agents has systematically validated verbalized reward signals from LLMs, establishing them as a methodologically grounded solution for credit assignment [3,4,5,6].
>
> Building on these, our MAR process includes two principled mechanisms to enhance the rigor of the estimation process:
> - **Stable baseline:** MAR leverages the raw global frontier trajectories $\mathcal{F}$ as input contexts, as a proxy for $V(s)$. Unlike a fluctuating critic network, this provides a stable baseline of best achieved value so far.
> - **Multi-trajectory Aggregation:** By aggregating outcomes over multiple local trajectories, MAR identifies consistent advantages, distinguishing causal signals from spurious ones.
>
> Our theoretical analysis in Appendix A provides formal justification of how these mechanisms reduce variance. Finally, the empirical results in Table 3 support our claim, where increasing from n=1 (no MAR) to n=3 in multi-trajectory comparison consistently improves performance.
>
> **References**
>
> [1] Rewarding progress: Scaling automated process verifiers for LLM reasoning. Setlur et al., ICML 2025.
>
> [2] VinePPO: Refining credit assignment in RL training of LLMs. Kazemnejad et al., ICML 2025.
>
> [3] Reflexion: Language Agents with Verbal Reinforcement Learning. Shinn et al., NeurIPS 2023.
>
> [4] Agent-as-a-Judge: Evaluate Agents with Agents. Zhuge et al., arXiv 2025.
>
> [5] Reflective Multi-Agent Collaboration based on Large Language Models. Bo et al., NeurIPS 2024.
>
> [6] Policy Improvement using Language Feedback Models. Zhong et al., NeurIPS 2024.
>
> ---
>
> ### [w2] Detailed analysis of LLM calls and API costs
>
> We provide additional details of LLM API usage for all LLM-based methods below. We report the average token consumption and costs across 6 games (Zork1, Zork3, Deephome, Ludicorp, Detective, Temple). When using GPT-4.1-mini for all LLM components (\\$0.40/\\$1.60 per million input/output tokens), per-run costs of all LLM-based approaches with 1,000 environment steps ranged from \\$1 to \\$7, maintaining practicality for research iteration. When using the more performant but costlier GPT-4.1 (\\$2.00/\\$8.00 per million input/output tokens), per-run costs ranged from \\$7.5 to \\$45.00, demonstrating that while stronger models increase costs by 5\~6x, the relative efficiency of our approach remains consistent with ∼40% fewer tokens than ICRL, while achieving superior performance.
>
> **Comparison of LLM API costs with GPT-4.1-mini**
>
> | Method | Input tokens | Output tokens | Total tokens | Avg. cost/run |
> |--------|-------------|---------------|--------------|---------------|
> | ICRL | 17.6M | 43.1K | 17.6M | $7.10 |
> | **GLoW** | 10.9M | 115.1K | 11.0M | $4.54 |
> | Reflexion | 7.2M | 50.6K | 7.3M | $2.98 |
> | IGE | 3.9M | 44.1K | 3.9M | $1.61 |
> | ReAct | 2.6M | 35.5K | 2.6M | $1.08 |
>
>
> **Comparison of LLM API costs with GPT-4.1**
>
> | Method | Input tokens | Output tokens | Total tokens | Avg. cost/run |
> |--------|-------------|---------------|--------------|---------------|
> | ICRL | 22.4M | 55.1K | 22.5M | $45.24 |
> | **GLoW** | 13.3M | 126.2K | 13.4M | $27.58 |
> | Reflexion | 9.9M | 62.5K | 10.0M | $20.31 |
> | IGE | 4.9M | 64.2K | 5.0M | $10.33 |
> | ReAct | 3.5M | 58.2K | 3.5M | $7.45 |

---

> ### Author Response · Authors · 2025-11-23
> **Response to Reviewer pUbB (continued)**
>
> ### [q2] Qualitative analysis of failure modes
>
> Thank you for this valuable suggestion. We have conducted a detailed qualitative analysis identifying two primary failure modes observed in GLoW trajectories (GPT-4.1-mini):
> 1. **Conservative State Selection:** In Zork1, we observe that GLoW can repeatedly select early-game states (`State 37: Kitchen, score 15, empty inventory`) despite frontier states with significantly higher scores (e.g. `State 58: Chasm, score 45, Lantern and Sword in inventory`) being available. Analysis reveals the model over-prioritizes the acquisition of safety-associated items (`lantern`, `sword`) from the environment, failing to recognize frontier states already possess these items in inventory. We posit that this stems from two interacting factors: (a) logical reasoning failure to verify item possession in the frontier states, and (b) an overvaluation of safety-related semantics, which distorts the estimation of potential value ($v’$ in Eq.2 of the draft). This biased selection can create a loop (at least temporarily), where the agent re-solves completed objectives rather than progressing from the frontier.
> 2. **Multi-step Dependency Reasoning Failure:** In Deephome, we illustrate a case where exploration stalls despite unlocking a critical mechanism, due to a failure to resolve subsequent multi-step dependencies. After activating the `City Generator` (step 827, +30 points), which powers the `railway` system, the agent fails to pursue the unlocked paths. The solution requires using the powered `railway` to access intermediate areas (`Blacksmith`, `Waterfall`) containing prerequisites for further objectives `Water Works` (+30) and `City Gates` (+40). However, the agent attempts the `Water Works` objective directly (2 failed attempts) and reverts to exploring familiar areas, never testing the `Railway` (0 visits in the rest of the game). This reveals a difficulty in systematically resolving multi-step dependencies, particularly when many intermediate objectives are present.
>
> We present the two cases in detail below:
>
> | Case | **Case 1: Zork1** &nbsp;&nbsp;&nbsp;&nbsp;&nbsp;&nbsp;&nbsp;&nbsp;&nbsp;&nbsp;&nbsp;&nbsp;&nbsp;&nbsp;&nbsp;&nbsp;&nbsp;&nbsp;&nbsp;&nbsp;&nbsp;&nbsp;&nbsp;&nbsp;&nbsp;&nbsp;&nbsp;&nbsp;&nbsp;&nbsp;&nbsp;&nbsp;&nbsp;&nbsp;&nbsp;&nbsp;&nbsp;&nbsp;&nbsp;&nbsp;&nbsp;&nbsp;&nbsp;&nbsp;&nbsp;&nbsp;&nbsp;&nbsp;&nbsp;&nbsp;&nbsp;&nbsp;&nbsp;&nbsp;&nbsp;&nbsp;&nbsp;&nbsp;&nbsp;&nbsp;&nbsp;&nbsp;&nbsp;&nbsp;&nbsp;&nbsp;&nbsp;&nbsp;&nbsp;&nbsp;&nbsp;&nbsp;&nbsp;&nbsp;&nbsp; | **Case 2: Deephome** &nbsp;&nbsp;&nbsp;&nbsp;&nbsp;&nbsp;&nbsp;&nbsp;&nbsp;&nbsp;&nbsp;&nbsp;&nbsp;&nbsp;&nbsp;&nbsp;&nbsp;&nbsp;&nbsp;&nbsp;&nbsp;&nbsp;&nbsp;&nbsp;&nbsp;&nbsp;&nbsp;&nbsp;&nbsp;&nbsp;&nbsp;&nbsp;&nbsp;&nbsp;&nbsp;&nbsp;&nbsp;&nbsp;&nbsp;&nbsp;&nbsp;&nbsp;&nbsp;&nbsp;&nbsp;&nbsp;&nbsp;&nbsp;&nbsp;&nbsp;&nbsp;&nbsp;&nbsp;&nbsp;&nbsp;&nbsp;&nbsp;&nbsp;&nbsp;&nbsp;&nbsp;&nbsp;&nbsp;&nbsp;&nbsp;&nbsp;&nbsp;&nbsp;&nbsp;&nbsp;&nbsp;&nbsp;&nbsp;&nbsp;&nbsp; |
> | :--- | :--- | :--- |
> | **Game Context** | **Game progress:** | **Game progress:** |
> | | • Reached deeper frontier states through Troll Room to **Maze (score 50)** and **Chasm (score 45)**. | • Agent has activated **City Generator (+30 points)** which powers Railway Station controls. |
> | | | • Progress requires using the **Railway** to access further intermediate areas (Blacksmith, Waterfall). |
> | | **State Archive:** | |
> | | • [State 72] Score: 50, Location: Maze, Inventory: lantern, sword... | |
> | | • [State 58] Score: 45, Location: Chasm, Inventory: lantern, sword | |
> | | • ... | |
> | | • [State 37] Score 15, Location: Kitchen, Inventory: **empty** | |
> | **Observed Behavior** | **Repetitive state selection:** | **Failed subsequent multi-stage progression:** |
> | | • Iter. 2: [State 37] (score 15) | • Step 827: Activates generator (+30 points, score 65) |
> | | • Iter. 3: [State 37] (score 15) | • Steps 828-1000: |
> | | • Iter. 4: [State 37] (score 15) | &nbsp;&nbsp;– Railway Station visits: **0** |
> | | ... | &nbsp;&nbsp;– Blacksmith visits: **0** |
> | | **Reasoning for state selection:** | &nbsp;&nbsp;– Waterfall visits: **0** |
> | | *"State 37 is in the living room where the brass lantern and the elvish sword are present. **Acquiring and activating these items is crucial for safe exploration**... This state aligns well with investigation goals..."* | &nbsp;&nbsp;– Water Works: 2 attempts at wheel, both fail |

---

> ### Author Response · Authors · 2025-11-23
> **Response to Reviewer pUbB (continued)**
>
> ### [w3] Justification for the choice of Jericho environment
>
> Thank you for this important question. We chose Jericho over IGE’s environments for the following principled reasons:
> - **Established benchmark suite covering diverse challenges:** Jericho has been leveraged as a self-contained benchmark by several prior works, including KG-A2C [1], MC-LAVE [2], XTX [3] and MC-DML [4], all of which evaluate exclusively on Jericho. This is due to the wide range of games which are commonly evaluated, each representing a unique combination of challenges including navigation, object manipulation and inventory management, and puzzle solving. Evaluation on 10 diverse games in the Jericho suite ensures methods are not overfitting on specific tasks, and allows us to perform a direct comparison against abovementioned baselines.
> - **Combinatorial complexity with fewer artificial constraints:** While IGE’s environments provide valuable controlled settings, they feature bounded complexity with explicit task instructions. Game of 24 involves finding basic arithmetic permutations of 4 numbers between 1 and 13, BabyAI consists of achieving one specified target configuration (e.g. “Put the red key next to a yellow box”) in a 6x6 2D grid, and the Cooking Game in TextWorld involves locating up to 5 ingredients from 13 rooms and processing/cooking food items, based on a provided recipe. In contrast, Jericho games feature massive combinatorial state and action spaces. For example, Zork I contains 110 rooms (Maintenance Room, Reservoir, Forest Path, Narrow Passage, Attic etc.), and 70 interactable objects (sword, lantern, treasures, lunch, control panel, key, etc.). And unlike games which provide a simple target objective, Jericho games require agents to explore to discover intermediate goals, as well as hidden causal dependencies required to solve those goals.
>
> These differences in scale and complexity of exploration make the Jericho games appropriate for evaluating our approach, which addresses these challenges through a global world model for navigating massive state spaces, and local semantic advantage for guiding exploration without instructions.
>
> **References**
>
> [1] Graph Constrained Reinforcement Learning for Natural Language Action Spaces. Ammanabrolu et al., ICLR 2020.
>
> [2] Monte-Carlo Planning and Learning with Language Action Value Estimates. Jang et al., ICLR 2021.
>
> [3] Multi-Stage Episodic Control for Strategic Exploration in Text Games. Tuyls et al., ICLR 2022.
>
> [4] Monte Carlo Planning with Large Language Model for Text-Based Game Agents. Shi et al., ICLR 2025.
>
> ---
>
> ### [w4] Reference formatting
> We appreciate your bringing this issue to our attention. We will ensure that references are formatted consistently in the revised draft.
>
> ---
>
> ### [q4] Potential applications in LLM reasoning and exploration ability of RLVR algorithms.
>
> Thank you for this interesting question. We believe there is significant potential for adapting GLoW for improving exploration in RLVR algorithms, by offering a structured alternative to existing statistical exploration methods in RLVR.
>
> Recent works [1,2,3] have identified “entropy collapse” as a significant bottleneck in RLVR, where models tend to trade exploratory uncertainty (quantified by policy entropy) for reward too quickly, converging on narrow behaviors and losing the incentive to explore alternative strategies. This lack of exploration weakens the model’s capacity for sustained, multi-step reasoning, causing performance to plateau [1].
>
> As a solution, existing approaches have employed statistical exploration methods drawn from RL literature, such as restricting exploitation by limiting updates on high-covariance tokens [1], or incentivizing uncertainty via perplexity-based curiosity bonuses [2] and entropy-based bonuses [3].
>
> Applying GLoW to RLVR would offer a structural alternative to these statistical solutions. By maintaining an explicit “reasoning frontier” of valid partial reasoning paths, GLoW can allow the LLM to systematically return to high-potential reasoning branch points, which would alleviate the problem caused by entropy collapse. Furthermore, for exploring reasoning paths, semantic advantages from MAR can provide dense intermediate reward signals based on evaluating multiple reasoning paths. We believe that this would be a promising shift that can unlock deeper exploration in RLVR.
>
> **References**
>
> [1] The Entropy Mechanism of Reinforcement Learning for Reasoning Language Models. Cui et al., arXiv 2025.
>
> [2] CDE: Curiosity-Driven Exploration for Efficient Reinforcement Learning in Large Language Models. Dai et al., arXiv 2025.
>
> [3]  Reasoning with Exploration: An Entropy Perspective. Cheng et al., AAAI 2026.

---

### Official Review · Reviewer_XV8D · 2025-10-31

**Soundness:** 3
**Presentation:** 1
**Contribution:** 2
**Rating:** 4
**Confidence:** 5

**Summary:**

This paper introduces GLoW, an LLM agent framework for tackling hard-exploration problems through the combination of a global and local world model. The global world model keeps track of a trajectory frontier of states to return back to, while the local world model performs Multi-path Advantage Reflection (MAR). The authors evaluate their method on the Jericho benchmark, a well-established benchmark for text-based games, and find their method to be state-of-the-art among LLM-based approaches (but not among pure RL approaches). Finally, the authors also provide ablation studies to see how different components of their method contribute to the final performance.

**Strengths:**

1. The authors provide strong empirical results: they find their method to be state-of-the-art among LLM-based approaches. While they are still a bit behind the strongest RL-based approaches, the proposed method uses a lot less environment steps.
2. Conceptually, I like the authors' proposal to think about *semantic* optimism to describe exploration potential, despite being only loosely implemented in prompts.
3. I commend the authors for providing a data contamination analysis in Appendix C.
4. I like Algorithm 3 in the appendix, which shows the differences and similarities between various Go-Explore-based algorithms.

**Weaknesses:**

1. Why is Algorithm 1 in the main text the Go-Explore Algorithm? This feels a bit strange as this is an already existing algorithm and is nothing new.
2. Page 3: “…. preserving complete trajectories enables accurate credit assignment and value estimation in sparse-reward environments where success depends on precise action sequences” → Could the authors explain why this is the case? There seems to be no evidence or reasoning provided for this.
3. [Minor] Page 3: this sentence needs rewriting: “This is an effective choice for Jericho’s sparse reward structure, and the possibility of encountering negative rewards or terminal failures.”
4. Page 3: “to extract a set of critical global states with value annotations” → what are “critical” global states?
5. Page 3: “Once a state is chosen, we replay the stored sequence of actions to return to the state” → does this mean the method only works in the deterministic setting? If so, I think the authors should mention this as a limitation since some of the prior works (e.g. DRRN, XTX) can operate in both settings.
6. Page 5: how is the actual advantage identification that’s visualized in Figure 2 done? It doesn’t seem to be described as far as I could see. From the appendix it seems it's all done via prompting, but that's not made clear in the main text.
7. Page 9: “While traditional world models in model-based RL focused on transition dynamics (Ha & Schmidhuber, 2018; Hafner et al., 2024), recent works show that in the context of LLMs, world models are usefully expanded as mechanisms for extracting task-sufficient state representations (Tang et al., 2024; Li et al., 2024).” → I disagree with the authors here that the current work can really be thought of as world models, which are usually grounded in transition models (even in the mentioned work of Tang et al., 2024). The current work does not build any such thing, and hence I would recommend the authors change the phrasing from world model to something else that more accurately depicts the proposed algorithm.
8. The writing is somewhat weak in its presentation and contains quite a few places with spelling or grammatical errors.
9. Figure 1: what does align_{LLM} do? It doesn’t seem to be explained.
10. Method seems to require quite a bit of hand crafted prompt design based on the prompts in Appendix D.
11. Since some of the standard errors are a bit large (e.g. 20.1 for Enchanter), it would be good to run a few more seeds.

**Questions:**

n/a

---

> ### Author Response · Authors · 2025-11-23
> **Response to Reviewer XV8D**
>
> Dear reviewer XV8D,
>
> We greatly appreciate your careful review of our paper and thoughtful feedback. We are glad that you found our method to be empirically strong, with state-of-the-art LLM results with higher sample efficiency compared to strong RL methods. We appreciate your concerns and valuable questions, and provide detailed responses to each below:
>
> ---
>
> ### [w1, w3, w8] Presentation and writing quality
>
> We appreciate your bringing these issues to our attention. We have conducted a thorough proofreading of the draft to enhance the presentation and writing quality. We will shortly upload the revised draft with a followup comment, which addresses these points to ensure clarity and precision.
>
> ---
>
> ### [w2] Why do complete trajectories enable credit assignment?
>
> Thank you for this great question. Full trajectories enhance credit assignment in Jericho, as progress often requires precise sequences of actions which are associated only with sparse rewards. For instance, in Zork1 the sequence [`Take lantern` → `Take sword` → `Go down into the Cellar` → `Reach Troll Room` → `Defeat Troll with Sword`] is required to progress further in the game, but only one of these actions (Go down into Cellar) has an associated reward. This makes credit assignment challenging with traditional RL algorithms, and methods like DQN rely on repeated trial and error with many rollouts to discover this sequence.
>
> In contrast, by providing LLM with complete trajectories from multiple exploration attempts, GLoW can infer credit assignment across trajectories. For instance, one trajectory without a lit `Lantern` cannot explore the dark `Cellar`, while another with `Lantern` but no `Sword` will successfully enter the `Cellar`, but die to the `Troll`. By analyzing both trajectories together, the global world model will identify that both items are needed to defeat the Troll, enabling accurate value estimation for states possessing these items, despite sparse rewards.
>
> ---
>
> ### [w4] What are critical global states?
>
> We clarify that critical global states are the state abstractions at the global level identified by $G_{llm}$ as semantic landmarks (e.g. frontiers, bottlenecks, and milestones). Operationally, they are the discrete entries in the analysis text $W_{global}$ generated by $G_{llm}$.
>
> ---
>
> ### [w5] Deterministic assumption and limitations
>
> You are correct that our implementation of action replay assumes a deterministic setting. We will make this limitation clear in the revised draft. Meanwhile, we note that our trajectory-based design facilitates intuitive extensions to handle stochasticity: A possible solution is to treat state restoration as a goal-reaching problem, and designing an LLM agent to return to the target state, guided by the replay trajectory as well as additional relevant trajectories as context.  We believe this is a promising direction for future work.

---

> ### Author Response · Authors · 2025-11-23
> **Response to Reviewer XV8D (continued)**
>
> ### [w6] How is the actual advantage identification performed?
>
> MAR is an LLM operation formally defined as $W_{local}\leftarrow\textrm{MAR}(\mathcal{T}_{s},\mathcal{F})$ (L226 of the draft), where $W\_{local}$ is a textual output which encodes semantic advantages for a set of critical states $s_1^\*, ..., s_k^\*$.
>
> We clarify that while MAR is LLM-based, it features a principled design for semantic approximation of the advantage function $A(s,a) = Q(s,a) - V(s)$:
> - It incorporates the raw global frontier trajectories $\mathcal{F}$ as input contexts, acting as a stable value baseline $V(s)$ driven by the knowledge of the current exploration frontier.
> - By focusing advantage estimation to critical states $s\_1^\*, ..., s\_k^\*$, it can aggregate and abstract information from multiple trajectories $\mathcal{T}\_s = \\{\tau\_1, ..., \tau\_n\\}$, linking key states (i.e. nodes in Fig.2) with their identifiable advantages.
>
> These mechanisms enable GLoW to bypass the challenge of numerical Q-value estimation in sparse reward settings, and accurately estimate advantages for key states through LLM reasoning. We will make this point clearer in the revised draft.
>
> Below, we show an abbreviated example of $W_{local}$ generated by MAR, demonstrating extracted states and advantages (The full example is available in Appendix E.2).
>
> | KEY STATES | ADVANTAGES |
> | :--- | :--- |
> | **The Troll Room** *(troll blocks exits; sword glows)* | • **"hit troll with sword"** $\rightarrow$ kills troll, clears all exits for exploration (score: 40) |
> | | • **"take axe"** after troll death $\rightarrow$ obtains valuable weapon for later puzzles |
> | | • **avoid engaging without glowing sword** $\rightarrow$ likely failure |
> | **Maintenance Room** *(colored buttons; leaking pipe)* | • **"push blue button"** $\rightarrow$ triggers leak, raises water level to access new areas |
> | | • **"take wrench, screwdriver, tube"** $\rightarrow$ essential tools for environment interaction |
> | | • **avoid throwing lantern** $\rightarrow$ breaks critical light source |
> | **Temple / Torch Room** *(dome; rope; altar; bell)* | • **"take ivory torch"** $\rightarrow$ stable light for deeper cave exploration |
> | | • **"ring bell at Entrance to Hades"** $\rightarrow$ paralyzes wraiths, enables passage |
> | | • **optimal:** acquire torch $\rightarrow$ bell $\rightarrow$ sceptre $\rightarrow$ manipulate altar |
> | **East-West Passage** *(chasm; railing; multiple paths)* | • **"tie rope to railing"** $\rightarrow$ enables safe descent into lower levels |
> | | • **"east" then "north"** $\rightarrow$ leads to Reservoir South and further areas |
> | | • **avoid getting stuck in loops** $\rightarrow$ wastes moves |
>
> ---
>
> ### [w7] World Model Terminology and Positioning
>
> Thank you for raising this important point. Please find our detailed discussion in Common Response 2, where we clarify that while our method does not learn a predictive dynamics model $P(s'|s,a)$, it aligns well with the "implicit representation" paradigm of world model, supported by recent literature on goal-oriented state abstractions in LLMs. We respectfully argue that GLoW’s structured abstractions at dual scales fulfill the functional role of a world model in decision-making without full dynamics simulation.
>
> ---
>
> ### [w9] Details of $align_{LLM}$
> $align_{LLM}$ performs LLM-based state selection, taking $W_{global}$ and the state archive as input, and outputting the chosen state $s_{next}$, shown in the state selection procedure in L171~179. Its goal is to infer the alignment between each archive state $s \in \mathcal{A}$ and $W_{global}$. For efficiency, we process all states at once, prompting the LLM to output a selected state index. We show the prompt for $align_{LLM}$ in Appendix D.2.
>
> ---
>
> ### [w10] Method requires hand crafted prompt design
> We appreciate this concern, and clarify that while we leverage hand-crafted prompts, we use the same set of zero-shot prompts across all games and experiments (for both GPT-4.1-mini and GPT-4.1), indicating that GLoW’s architecture is general and robust. Furthermore, we leverage a simple ReAct-style prompt for the agent, and the entire framework requires only 3 additional prompts corresponding to each of the components, $g_{LLM}$, $align_{LLM}$, and $\text{MAR}$.
>
> ---
>
> ### [w11] Larger standard errors on some games
> Thank you for this suggestion. We conducted 2 more runs for Enchanter, obtaining scores of [45, 90, 50, 85, 45] with mean 63.0 and standard deviation 20.1. The variance arises from Enchanter's reward structure itself, where agents either solve major discovery-based puzzles in the castle interior with large rewards (e.g. +25) leading to 85\~90 scores, or plateau at an earlier exploration frontier (scores 45\~50). This pattern reflects the hard-exploration nature of the task, rather than instability of the framework.

---

> ### Comment · Reviewer_XV8D · 2025-11-26
>
> I thank the authors for the rebuttal - I especially appreciate the clarification on the generality of the hand-crafted prompt design across games and models. Also, I'm impressed by the new numbers the authors reported on the full 4.1 model.
>
> My main remaining concern is that the authors often claim their method is implementing approximations of precise concepts from RL such as advantages or value baselines, while I do not see any formal connections between them. A few examples:
> - *"We clarify that while MAR is LLM-based, it features a principled design for semantic approximation of the advantage function "* MAR outputs text, while the advantage function outputs scalar values. I do not see how MAR is providing an approximation of the advantage function in any formal sense.
> - *"It incorporates the raw global frontier trajectories as input contexts, acting as a stable value baseline driven by the knowledge of the current exploration frontier."* I do not see the analogy between the frontier trajectories and a stable value baseline V(s).
> - "In Appendix A, we provide theoretical analysis showing how MAR reduces variance through both multi-trajectory comparison and the stable baseline based on F." MAR outputs text and, if I understand correctly, doesn't do any explicit averaging of multiple trajectory rollouts the way it is shown in the proof. Therefore, I find the statement "theoretical analysis showing how MAR reduces variance" to be misleading, as the proof is unrelated to the practical MAR procedure.
>
> A few additional minor comments:
> 1. I'd recommend the authors to report one standard error instead of one standard deviation. The standard error will shrink with more seeds but the standard deviation will not.
> 2. I still find it a bit strange the authors have the Go-explore algorithm without any modifications as Algorithm 1 in the main text of the paper, since this algorithm existed before and the algorithm box introduces nothing new. I'd recommend maybe augmenting it with the changes that the proposed algorithm makes to the Go-explore algorithm (or how it implements it differently).
> 3. In the experimental setup, the authors write they use 50-step episodes for ReAct and Reflexion. Is this enough for the agent to reach it's full potential, or does it sometimes truncate game play when the agent could have kept going?
> 4. For "world model", I'm sympathetic to the broadened interpretation the authors are going for in this work, but I still believe it would benefit the clarity of the work to maybe make this more explicit by replacing "world model" with something like "textual world model" or something that leads readers who are expecting a dynamics model less astray.

---

> > ### Author Response · Authors · 2025-11-27
> > **Response to Reviewer XV8D**
> >
> > Thank you for your constructive feedback. We are very pleased to hear that you found our GPT-4.1 results impressive, and appreciated our clarifications on prompt generality.
> >
> > To address your main concern, we clarify that we do not claim to implement precise numerical approximations of RL quantities. Rather, we have drawn from RL principles to design LLM-based mechanisms serving analogous functional roles in guiding LLM agent behavior.
> >
> > **[c1. How MAR relates to the numerical advantage function]**
> >
> > We agree that our rebuttal phrasing of "semantic approximation of the advantage function" may incorrectly suggest MAR performs a formal numerical approximation of $A(s,a)$, which it does not do. We apologize for this confusion.
> >
> > To clarify, MAR produces semantic advantages (structured textual representations identifying beneficial actions at key states), which serve a functional role analogous to advantages in guiding the LLM agent's exploration policy. Concretely, $W_{local}$ is provided as input context to $\pi_{explore}$ (Eq. 3, prompt in Appendix D.4), directly informing action selection. We validate this mechanism empirically through our MAR ablation in Table 2, showing that replacing MAR's semantic advantages with standard Reflexion degrades performance across games.
> >
> > **[c2. How frontier trajectories $\mathcal{F}$ relate to $V(s)$]**
> >
> > The frontier trajectories $\mathcal{F}$ serve an analogous role to the value baseline $V(s)$ within the MAR operation. As shown in the MAR prompt (Appendix D.3), the LLM receives the frontier trajectories (representing the best outcomes achieved so far) alongside local exploration trajectories. In this context, the trajectories in $\mathcal{F}$ provide a stable reference point for comparison, as the LLM evaluates whether new trajectories constitute progress relative to these known high-value outcomes. This implements the role of a value baseline through context-based LLM reasoning, rather than numerical subtraction.
> >
> > **[c3. Role of theoretical analysis (Appendix A)]**
> >
> > Thank you for raising this point. You are correct that the proof does not directly establish formal properties of MAR itself. The aim of Appendix A is to provide theoretically grounded motivation for why multi-trajectory comparison from the same state is beneficial. The analysis shows that aggregating information reduces variance in advantage estimation. With this motivation, MAR applies this principle through comparative LLM reasoning, instead of explicit numerical averaging. We will clarify in the revision that Appendix A motivates the design, while Table 3 (n=1 vs n=3) provides empirical validation that this principle benefits our implementation in practice.
> >
> > **[c4. Standard error]**
> >
> > Thank you for this suggestion. We will report standard error in the revised draft.
> >
> > **[c5. Presentation of Alg.1]**
> >
> > Thank you for this great suggestion. We will revise Alg.1 to show Go-Explore with GLoW's modifications highlighted, to make our contributions clearer.
> >
> > **[c6. 50-step episodes for baseline agents]**
> >
> > Thank you for this question. While it is possible that truncation happens, in our observation baseline agents typically plateau early, getting stuck on puzzles or falling into repetitive loops before the 50-step limit, making 50 steps sufficient to capture the full potential of baseline agents (ReAct, Reflexion, ICRL).
> >
> > **[c7. World model terminology]**
> >
> > We appreciate your acknowledgement of our broader definition. We agree that "textual world model" would reduce confusion, and will adopt this terminology to avoid confusion with dynamics world models. We appreciate this thoughtful suggestion.
> >
> > ---
> >
> > We will upload a revised PDF reflecting these changes before the discussion period ends. Thank you again for your valuable feedback, and we hope that our response addressed your remaining concerns. Please don't hesitate to let us know if you have any further questions.

---

> > ### Author Response · Authors · 2025-11-29
> > **Response to Reviewer XV8D**
> >
> > We have uploaded a revised PDF reflecting the discussion, with requested revisions noted below:
> >
> > **World Model Terminology**
> > - Adopted "textual world model" terminology throughout (L13-14, L37-40, L107, L468-475)
> >
> > **MAR framing w.r.t RL theory**
> > - Clarified functional analogy vs numerical approximation (L243-253, L268-269; Appendix A L710-719)
> > - Clarified frontier as stable reference point (L250-253; Appendix A L720-724)
> > - Refined framing of theoretical results as "Theoretical Motivation" (Appendix A L668-724)
> >
> > **Presentation Clarity**
> > - Algorithm 1 updated to highlight GLoW's modifications (L99-102, L108-123)
> > - Sentence rewriting (L136)
> > - Complete trajectories explanation added (L139-143)
> > - "Critical global states" clarified (L162-163)
> > - alignLLM explanation added (L177)
> > - Clarified 50-step episode sufficiency for baselines (L326-327)
> > - Clarified standard deviation usage for consistency with baselines (L352)
> > - Thorough proofreading and grammatical errors fixed
> >
> > **Limitations**
> > - Noted deterministic assumption (L215; Appendix F L1201-1210)
> >
> > We believe the rebuttal and revised PDF fully address the concerns raised. Thank you again for your thorough and constructive feedback, which meaningfully improved our paper.

---

### Official Review · Reviewer_Ru4g · 2025-11-02

**Soundness:** 3
**Presentation:** 3
**Contribution:** 3
**Rating:** 6
**Confidence:** 4

**Summary:**

The paper introduces GLoW, a dual-scale framework designed to improve exploration efficiency of LLM-based agents in hard-exploration environments such as the Jericho benchmark. It combines a Global World Model, which maintains a trajectory frontier and uses LLM reasoning to estimate both achieved and potential state values for principled state selection, with a Local World Model, which applies Multi-path Advantage Reflection (MAR) to infer semantic advantage signals from multiple rollouts under sparse rewards. Experiments show that GLoW achieves new state-of-the-art performance among LLM agents.

**Strengths:**

- The proposed framework achieves strong empirical performance and high sample efficiency on multiple games from benchmark.
- The proposed semantic advantage representation enhances interpretability compared to scalar rewards.

**Weaknesses:**

Overall, I like this paper, I just have a few minor questions:
In the global world model, how exactly is a trajectory defined? If an episode does not reach a terminal state (e.g., success or failure), does it default to a fixed-length rollout of 50 steps?
Would excessively long interactions degrade the reasoning quality of the LLM? Conversely, if the LLM is capable of longer reasoning chains, can the trajectory length be adaptively extended? Simply truncating trajectories might risk losing key contextual information.

**Questions:**

The paper mentions using GPT-4.1-mini for trajectory analysis. I wonder whether smaller models could achieve comparable effects. It would be insightful to include a few case studies exploring how different model influences the quality of world-model reasoning and trajectory evaluation.

---

> ### Author Response · Authors · 2025-11-23
> **Response to Reviewer Ru4g**
>
> Dear reviewer Ru4g,
>
> Thank you very much for your thoughtful review and questions. We are pleased to hear that you found our method empirically strong and highly sample efficient, with an interpretable semantic advantage representation. We appreciate your valuable questions and comments, and provide detailed responses to each below:
>
> ---
>
> ### [w1] Trajectory definition and interaction length trade-offs
> Thank you for this insightful question. Each trajectory in the trajectory frontier processed by the global world model represents a 50-step exploration episode from a selected starting state. You are correct that if an episode does not reach a terminal state (victory or death), the trajectory is truncated at 50 steps.
>
> We chose this fixed length to balance context preservation with reasoning degradation. Your intuition is correct that longer interaction sequences can degrade the quality of LLM reasoning, in particular because the global world model must analyze the entire set of frontier trajectories simultaneously. In our preliminary tests, we found 50 steps to strike an effective balance, keeping the scope of each trajectory manageable for analysis.
>
> Regarding a potential loss of contextual information from truncation, we note that this is mitigated by the fact that the next exploration episode can start from any selected state from previous trajectories. As a result, we prioritized fixed lengths to ensure predictable token costs and consistent analysis scopes. Nevertheless, we believe dynamically extending the trajectory length is an interesting direction for future work, where a potential approach would be to use an LLM-based verifier to judge whether to extend the episode based on the trajectory context.
>
> ---
>
> ### [q1] Impact of Model Capability on World Model Quality
> We appreciate this valuable question. Based on your suggestion, to study the effect of different LLMs on world model and trajectory evaluation quality, we are currently conducting the requested additional analyses across GPT-4.1-nano, GPT-4.1-mini, and GPT-4.1 models. We will share the detailed results of these case studies shortly, with a followup comment.

---

> ### Author Response · Authors · 2025-11-25
> **Response to Reviewer Ru4g (continued)**
>
> ### [q1] Impact of Model Capability on World Model Quality
>
> In order to assess the impact of model capability on global world model and trajectory evaluation, we conducted a systematic comparison across three LLM model scales (GPT-4.1-nano, GPT-4.1-mini, and GPT-4.1), on two games (Zork1 and Deephome). To isolate the effect of model capability, we used an identical frontier trajectory set extracted from a GPT-4.1-mini checkpoint, generating a new $W_{global}$ for each LLM.
>
> The results are shown below. We present abbreviated trajectory contexts (the inputs for $W_{global}$ generation), then summarize each $W_{global}$ into content capturing achieved value ($v$) and potential value ($v'$).
>
> First, we observe that the evaluation of $v$ is consistent across model capabilities, indicating that even a smaller model (GPT-4.1-nano) can successfully evaluate trajectories to identify achieved goals and checkpoints.
> We present case study 1 in Zork1 (early game, troll puzzle) as the control case, where all three LLMs largely align in both $v$ and $v'$.
>
> In contrast, in other cases we observed clear qualitative differences, particularly in the generated $v'$, between the smaller GPT-4.1-nano and the larger models, GPT-4.1-mini and GPT-4.1. In case study 2 (Zork1 mid game), the trajectories present multiple simultaneous puzzles, and we find that the specificity of reasoning about these objectives shows qualitative differences correlating with model capability. GPT-4.1 and GPT-4.1-mini targeted specific objectives, while nano generated generic targets. In case study 3 (Deephome early game), models must infer that Ember (a town mentioned in a note) is the location containing necessary items to proceed. This information must be recalled from reading a note that provides this locational clue. We find that while the larger models are able to do so, GPT-4.1-nano fails, resulting in a generic "correct parts for elevators (mentioned in note)" missing the crucial Ember reference.
>
> Since the textual $v'$ directly influences subsequent state selection, LLM capability gaps in GWM generation propagate to exploration performance. We note the average performance across both games reflects these qualitative differences:
> | Model | Zork1 | Deephome |
> | :--- | :--- | :--- |
> | **GPT-4.1-nano** | 43.0 ± 2.2 | 18.3 ± 5.3 |
> | **GPT-4.1-mini** | 73.0 ± 4.5 | 75.0 ± 8.7 |
> | **GPT-4.1** | 103.0 ± 6.8 | 114.7 ± 27.8 |
>
> ---
>
> ### Case Study 1: Zork1 Early Game
>
> **Trajectory Context**
> > - **Traj 1,2 (score 40):** house → cellar → defeat troll → maze
> > - **Traj 0 (score 15):** house exploration, no troll
>
> **Achieved Value (v):**
> > All models consistently identify *"Defeating the Troll"* as the key milestone.
>
> **Potential Value (v'):**
> > - **GPT-4.1-nano:** *"Explore maze, find keys, unlock grating"*
> > - **GPT-4.1-mini:** *"Navigate maze, find safe chasm crossing, unlock grating"*
>
> ---
>
> ### Case Study 2: Zork1 Mid Game
>
> **Trajectory Context**
> > - **Traj 22,23 (score 70):** dam controls → reservoir drains → take trunk → +15 pts
> > - **Traj 15,16,18 (score 55):** dam area, Loud Room (can't take platinum bar), no trunk
> > - **Key observations:** Dam puzzles, Loud Room/platinum bar, reservoir access
>
> **Achieved Value (v):**
> > All models consistently identify key milestones: *"dam controls", "reservoir access", "trunk acquisition"*
>
> **Potential Value (v'):**
> > - **GPT-4.1-nano:** *"Secure tools (wrench, screwdriver, key)", "Explore blocked passages", "Use collected artifacts"* ($\color{red}{\text{generic objectives}}$)
> > - **GPT-4.1-mini:** *"Find how to turn valve at Dam Base", "Explore grating under leaves in clearing", "Explore reservoir as water levels change"* ($\color{green}{\text{specific locations and puzzles}}$)
> > - **GPT-4.1:** *"Solve Loud Room/Platinum Bar puzzle", "Cross chasm to dome room", "Unlock trophy case"* ($\color{green}{\text{specific puzzles and objectives}}$)
>
> ---
>
> ### Case Study 3: Deephome Early Game
>
> **Trajectory Context**
> > - **Traj 2 (score 20):** take note → examine warning → explores to City Gates
> > - **Traj 1 (score 8):** take note → examine warning → say manaz → death
> > - **Traj 0 (score 1):** say manaz → immediate death (never reads note)
> > - **Key observation:** Note text reads *"I traveled south to a human town called **Ember** to procure the correct parts"*
>
> **Achieved Value (v):**
> > All models consistently identify key milestones: *"reaching City Gates", "discovering control panel", "reading the note"*
>
> **Potential Value (v'):**
> > - **GPT-4.1-nano:** *"correct parts for elevators (mentioned in note)"* ($\color{red}{\text{no destination specified}}$)
> > - **GPT-4.1-mini/GPT-4.1:** *"Obtain elevator parts from **Ember** (human town)"* ($\color{green}{\text{specific target location}}$)

---

### Author Response · Authors · 2025-11-23
**General Response**

Dear Reviewers and AC,

We greatly appreciate your insightful and constructive reviews, and are glad to hear your acknowledgement of the positive aspects of our paper, including:
* **Novel dual-scale framework** with clear motivation and interesting conceptual contributions (XV8D, pUbB, PRB8)
* **Effective method using LLM reasoning** to infer dense semantic advantage signals from sparse rewards (Ru4g, pUbB, PRB8)
* **State-of-the-art performance** among LLM approaches, with much higher sample efficiency compared to RL approaches (Ru4g, XV8D, pUbB, PRB8)

In response to your feedback, we have conducted additional experiments and analyses which we detail in individual responses. In addition, below we address common questions from reviewers regarding the robustness of our method, and our usage of the world model terminology.

---

### [Common response 1: Robustness to LLM Capability (Ru4g, pUbB, PRB8)]
To address concerns regarding the robustness of GLoW to different LLMs , we evaluated GLoW with GPT-4.1 on the 6 Extreme/Difficult games. The results below show that with a stronger LLM, our method scales effectively, achieving the **highest performance on 4 out of 6 games** across all methods, and surpassing the SOTA RL baseline XTX on **5 out of 6 games**. These results validate the robustness of our proposed dual-scale approach, and we thank the reviewers for this valuable suggestion.

| Methods | RL | **GPT-4.1 mini** | | | | | **GPT-4.1** | | | | |
|-------|----------|---------|------|------|------|------|---------|------|------|------|------|
| |**XTX** | **ReAct** | **Reflexion** | **ICRL** | **IGE** | **GLoW** | **ReAct** | **Rfl** | **ICRL** | **IGE** | **GLoW** |
| **Steps** | 800K | 1K | 1K | 1K | 1K | 1K | 1K | 1K | 1K | 1K | 1K |
| **Enchanter** | 52.0 | 46.7$_{\pm 9.4}$ | 48.3$_{\pm 9.4}$ | 43.3$_{\pm 8.5}$ | 50.0$_{\pm 7.1}$ | 61.7$_{\pm 20.1}$ | 38.3$_{\pm 2.4}$ | 58.3$_{\pm 2.4}$ | 45$_{\pm 7.1}$ | $\underline{\mathsf{68.3}_{\pm 18.4}}$ | **98.3**$_{\pm 4.7}$ |
| **Zork1** | **103.4**$_{\pm 10.9}$ | 48.3$_{\pm 4.7}$ | 48.0$_{\pm 5.0}$ | 51.7$_{\pm 4.7}$ | 44.3$_{\pm 0.5}$ | 73.0$_{\pm 4.5}$ | 45.0$_{\pm 0.0}$ | 54.3$_{\pm 4.5}$ | 48.0$_{\pm 2.8}$ | 86.7$_{\pm 24.1}$ | $\underline{\mathsf{103.0}_{\pm 6.8}}$ |
| **Zork3** | 4.2$_{\pm 0.1}$ | 3.0$_{\pm 0.0}$ | 2.7$_{\pm 0.5}$ | 3.0$_{\pm 0.0}$ | 3.7$_{\pm 0.9}$ | $\underline{\mathsf{4.3}_{\pm 0.9}}$ | 3.3$_{\pm 0.5}$ | 2.7$_{\pm 0.5}$ | 3.0$_{\pm 0.8}$ | 3.0$_{\pm 0.0}$ | **5.0**$_{\pm 0.0}$ |
| **Deephome** | 77.7$_{\pm 2.1}$ | 11.0$_{\pm 4.2}$ | 22.0$_{\pm 1.6}$ | 24.0$_{\pm 5.7}$ | 71.3$_{\pm 4.9}$ | 75.0$_{\pm 8.7}$ | 32.3$_{\pm 19.6}$ | 22.3$_{\pm 1.7}$ | 34.7$_{\pm 18.7}$ | $\underline{\mathsf{82.0}_{\pm 8.6}}$ | **114.7**$_{\pm 27.8}$ |
| **Ludicorp** | 78.8 | 19.7$_{\pm 0.9}$ | 21.7$_{\pm 1.2}$ | 32.0$_{\pm 7.1}$ | 28.3$_{\pm 11.3}$ | 73.7$_{\pm 11.0}$ | 31.0$_{\pm 2.8}$ | 29.0$_{\pm 0.8}$ | 31.7$_{\pm 0.5}$ | **89.0**$_{\pm 7.8}$ | $\underline{\mathsf{79.0}_{\pm 16.8}}$ |
| **Balances** | $\underline{\mathsf{24}}$ | 10$_{\pm 0.0}$ | 10$_{\pm 0.0}$ | 11.7$_{\pm 2.4}$ | 10.0$_{\pm 0.0}$ | 16.7$_{\pm 2.4}$ | 18.3$_{\pm 2.4}$ | 18.3$_{\pm 2.4}$ | 16.7$_{\pm 2.4}$ | 16.7$_{\pm 2.4}$ | **26.7**$_{\pm 2.4}$ |

---

### [Common response 2: World Model Terminology and Positioning (XV8D, PRB8)]
We appreciate the reviewers for raising this important distinction. While our usage of the world model terminology differs from traditional model-based RL (learning predictive dynamics $P(s’|s,a)$), it aligns with an emerging LLM paradigm recognizing world models as mechanisms for “implicit representations of the external world” rather than strictly only as simulators.

Recent survey [1] categorizes world models into two branches: 1) Implicit representations (internalizing world knowledge for decision-making), and 2) Future prediction (simulating dynamics). While the traditional MBRL definition fits the latter, [1] validates the former: such models "abstract the world into simplified elements" to support downstream planning  without a full depiction of the world state. Similarly [2] formalizes this via state abstraction theory in RL [3], showing that effective LLM agents build goal-oriented abstractions guiding task completion without recovering full dynamics.

As noted by reviewer PRB8, by constructing "structured representations of experiential knowledge", our method aligns with this view. GLoW’s global and local world models implement goal-oriented state abstraction at dual scales, enabling principled state selection at the global scale, and guiding exploration under sparse rewards through local semantic advantages.

**References**

[1] Understanding World or Predicting Future? A Comprehensive Survey of World Models. Ding et al., ACM Computing Surveys, 2025.

[2] Do LLMs Build World Representations? Probing Through the Lens of State Abstraction. Li et al., NeurIPS 2024.

[3] A theory of abstraction in reinforcement learning. Abel, arXiv, 2022.

---

### Author Response · Authors · 2025-11-29
**Summary of Rebuttal**

Dear AC,

We greatly appreciate your time and effort in reviewing our paper. We would also like to thank all reviewers again for their valuable and constructive feedback. Below, we summarize the key issues and resolutions from the discussion period.

---

### Key Issues & Resolutions

**1. World model terminology (XV8D, PRB8)**

Two reviewers raised concern that "world model" has a classical MBRL definition of learning a dynamics model of $P(s'|s,a)$. We addressed this in our Common Response 2: Our definition aligns with an accepted broader view in recent LLM literature, which recognizes world models as mechanisms for "implicit representations of the external world" rather than strictly simulators. As reviewer PRB8 noted, GLoW constructs "structured representations of experiential knowledge" at global and local scales, which aligns with this view.

**Reviewer XV8D accepted our framing in their response:** *"I'm sympathetic to the broadened interpretation the authors are going for in this work"*, and suggested "textual world model" terminology as a clarifying distinction from dynamics models. We reflected this distinction in the revised PDF by adopting "textual world model" terminology (L13-14, L37-40, L107, L468-475).

**2. LLM robustness (Ru4g, pUbB, PRB8)**

Three reviewers raised concern about GLoW's robustness across LLMs. Reviewer PRB8 noted that while standardizing on GPT-4.1-mini provides clear comparison, *"a targeted ablation study, even on a subset of the Jericho games with a different LLM, would greatly enhance the paper's contribution"* by providing *"evidence that the dual-scale mechanism is a robust and generalizable principle."*

We addressed this directly by evaluating GLoW with GPT-4.1 on 6 Extreme/Difficult games (Common Response 1; Appendix G). The new results confirmed GLoW is robust across LLMs, and achieved **SOTA across all methods on 4/6 games**, surpassing XTX, the strongest RL baseline which uses 800K steps, on 5/6 games.

**These results were acknowledged by reviewer XV8D in their response:** *"I'm impressed by the new numbers the authors reported on the full 4.1 model."*

**3. MAR framing w.r.t. RL theory (XV8D)**

Reviewer XV8D raised concern that in the original draft, the formulation of theoretical results in Appendix A suggested MAR performs formal numerical approximation of advantages. In our follow-up response to reviewer XV8D, we clarified that MAR does not numerically approximate the advantage function, but is *motivated by* RL advantage estimation principles, operationalizing them through LLM reasoning.

**We reflected the discussion through the following revisions in the updated PDF:**
- We refined the framing of theoretical results as theoretical motivation (Appendix A L668-724)
- We clarified that MAR's multi-trajectory comparison provides a functional analogy to variance reduction, without numerical averaging (L243-253, L268-269; Appendix A L710-719)
- We clarified that trajectory frontier has an analogous functional role as value baseline (L250-253; Appendix A L720-724)

---

### Additional Reviewer Context
Since we were not able to engage further before the end of discussions, we would like to further highlight:
- **Reviewer Ru4g:** Stated *"I like this paper, I just have a few minor questions"*. We provided detailed responses to all questions, including a new case study on world model quality across GPT-4.1-nano/GPT-4.1-mini/GPT-4.1.
- **Reviewer pUbB:** Stated *"I am willing to update my evaluation if my questions and concerns are well addressed during the discussion stage"*. We responded to all questions and concerns raised, including providing additional detailed API cost analysis and qualitative analysis of failure modes.

---

### Revised PDF
We have uploaded the revised PDF referenced above on Nov. 29th, with revision details posted in our latest response to reviewer XV8D.

---

### Meta-Review · Area_Chair_WZAR · 2025-12-26

**Summary:**

Paper proposes a Go-Explore variant for text-based games. In their algorithm, GloW, LLMs with specific prompts replace the scalar reward/value based components in Go-Explore with semantic counterparts. The connection between the formal RL terms and their semantic counterparts, while currently a bit hand wavy (as mentioned by one reviewer) is very interesting, and may have impact beyond this specific algorithm/problem.

**Reviewer Concerns:**

The rebuttal addressed most reviewers concerns.

**World models positioning** - I agree with reviewers XV8D,PRB8 that this positioning is confusing, as at present, world models usually have a very different connotation. Compared to the other RL terms from Go-Explore and their elegant semantic counterparts, I didn’t find the world model terminology to add much to the paper, and **I would suggest to drop it**. If the authors insist on keeping this terminology, I suggest to at least explain it well very early in the paper.

**Reviewer Scores:**

6,4->6/8,4->?,6

This is a good paper. Decision based also on my own review of the work.

---

### Decision · Program_Chairs · 2026-01-26

Accept (Poster)